# HOW TO IMPROVE SAMPLE COMPLEXITY OF SGD OVER HIGHLY DEPENDENT DATA?

## ABSTRACT

Conventional machine learning applications typically assume that data samples are independently and identically distributed (i.i.d.). However, practical scenarios often involve a data-generating process that produces highly dependent data samples, which are known to heavily bias the stochastic optimization process and slow down the convergence of learning. In this paper, we conduct a fundamental study on how different structures of stochastic update schemes affect the sample complexity of stochastic gradient descent (SGD) over highly dependent data. Specifically, with a $\phi$-mixing model of data dependence, we show that SGD with proper periodic data-subsampling achieves an improved sample complexity over the standard SGD in the full spectrum of the data dependence level. Interestingly, even subsampling a subset of data samples can accelerate the convergence of SGD over highly dependent data. Moreover, we show that mini-batch SGD can further substantially improve the sample complexity over SGD with periodic data-subsampling over highly dependent data. We also conduct some numerical experiments to validate our theoretical results.

## 1 INTRODUCTION

Stochastic optimization algorithms have attracted great attention in the past decade due to its successful applications to a broad research areas, including deep learning (Goodfellow et al., 2016), reinforcement learning (Sutton & Barto, 2018), online learning (Bottou, 2010; Hazan, 2017), control (Marti, 2017), etc. In the conventional analysis of stochastic optimization algorithms, it is usually assumed that all data samples are independently and identically distributed (i.i.d.) and queried. For example, data samples in the traditional empirical risk minimization framework are assumed to be queried independently from the underlying data distribution, while data samples in reinforcement learning are assumed to be queried from the stationary distribution of the underlying Markov chain.

Although the i.i.d. data assumption leads to a comprehensive understanding of the statistical limit and computation complexity of SGD, it violates the nature of many practical data-generating stochastic processes, which generate highly correlated samples that depend on the history. In fact, dependent data can be found almost everywhere, e.g., daily stock price, weather/climate data, state transitions in Markov chains, etc. To understand the impact of data dependence on the convergence and complexity of stochastic algorithms, there is a growing number of recent works that introduce various definitions to quantify data dependence. Specifically, to analyze the finite-time convergence of various stochastic reinforcement learning algorithms, recent studies assume that the dependent samples queried from the Markov decision process satisfy a geometric mixing property (Dalal et al., 2018; Zou et al., 2019; Xu & Gu, 2020; Qu & Wierman, 2020), which requires the underlying Markov chain to be uniformly ergodic or has a finite mixing time (Even-Dar et al., 2003). On the other hand, to analyze the convergence of stochastic optimization algorithms over dependent data, Karimi et al. (2019) assumed the existence of a solution to the Poisson equation associated with the underlying Markov chain, which is a weaker condition than the uniform ergodic condition (Glynn & Meyn, 1996). Moreover, Agarwal & Duchi (2012) introduced a $\phi$-mixing property of the data-generating process that quantifies how fast the distribution of future data samples (conditioned on a fixed filtration) converges to the underlying stationary data distribution. In particular, the $\phi$-mixing property is more general than the previous two notions of data dependence (Douc et al., 2018).

While the aforementioned works leveraged the above notions of data dependence to characterize the sample complexity of various standard stochastic algorithms over dependent data, there still lacks theoretical understanding of how algorithm structure affects the sample complexity of stochastic algorithms under different levels of data dependence. In particular, a key algorithm structure is the stochastic update scheme, which critically affects the bias and variance of the stochastic optimization process. In fact, under i.i.d. data and convex geometry, it is well known that SGD achieves the sample complexity lower bound under various stochastic update schemes (Bottou, 2010), e.g., single-sample update and mini-batch update. However, these stochastic update schemes may lead to substantially different convergence behaviors over highly dependent data, as they are no longer unbiased. Therefore, it is of vital importance to understand the interplay among data dependence, structure of stochastic update and convergence rate of stochastic algorithms, and we want to ask the following fundamental question.

- **Q:** How does the structure of stochastic updates affect the convergence rate and sample complexity of stochastic algorithms over dependent data?

In this paper, we provide comprehensive answers to the above fundamental question. Specifically, we conduct a comprehensive study of the convergence rate and sample complexity of the SGD algorithm over a wide spectrum of data dependence levels under various types of stochastic updates, including periodic subsampling and mini-batch sampling. Our results show that SGD with both stochastic updates achieves a substantially improved sample complexity over the standard SGD under highly dependent data. We summarize our contributions as follows.

## 1.1 OUR CONTRIBUTIONS

We consider the following standard stochastic optimization problem.

$$\min_{w \in \mathcal{W}} f(w) := \mathbb{E}_{\xi \sim \mu} \left[ F(w; \xi) \right], \tag{P}$$

where the objective function $f$ is convex and Lipschitz continuous, and the expectation is taken over the stationary distribution $\mu$ of the underlying data-generating process $\mathbf{P}$. To perform stochastic optimization, we query a stream of dependent data samples from the underlying data-generating process. Specifically, we adopt the $\phi$-mixing model to quantify the data dependence via a decaying mixing coefficient function $\phi_\xi(k)$ (see Definition 2.2) (Agarwal & Duchi, 2012). We study the convergence of the stochastic gradient descent (SGD) algorithm over $\phi$-mixing dependent data samples under various stochastic update schemes, including data subsampling and mini-batch sampling.

We first study the convergence of SGD over $\phi$-mixing dependent data samples under the data subsampling update scheme. In particular, the data subsampling update scheme utilizes only one data sample per $r$ consecutive data samples by periodically skipping $r - 1$ samples. With this data subsampling scheme, the subsampled data samples are less dependent for a larger subsampling period $r$. Consequently, we show that SGD with a proper data subsampling period achieves an improved sample complexity over that of the standard SGD in the full spectrum of the convergence rate of the $\phi$-mixing coefficient. In particular, the improvement is substantial when the data is highly dependent with an algebraic decaying $\phi$-mixing coefficient.

Moreover, we study the sample complexity of SGD over $\phi$-mixing dependent data samples under the mini-batch update scheme. Compare to the data subsampling update, mini-batch update can substantially reduce the mini-batch data dependence without skipping data samples. Consequently, mini-batch update leverages the sample average over a mini batch of data samples to reduce both the bias (caused by the data dependence) and the optimization variance. Specifically, we show that SGD with mini-batch update achieves an orderwise lower sample complexity than both the standard SGD and the SGD with data subsampling in the full spectrum of the convergence rate of the $\phi$-mixing coefficient. We summarize and compare the sample complexities of these stochastic algorithms under different $\phi$-mixing data dependence models in Table 1.

## 1.2 RELATED WORK

**Stochastic Algorithms over Dependent Data** Steinwart & Christmann (2009) and Modha & Masry (1996) established the convergence analysis of online stochastic algorithms for streaming

Table 1: Comparison of sample complexities of SGD, SGD with data subsampling and Mini-batch SGD under different levels of data dependence for achieving $f(w) - f(w^*) \leq \epsilon$. Note that $\theta$ is a parameter of the convergence rate of the $\phi$-mixing coefficient.

| Data dependence level | $\phi_\xi(k)$ | SGD | SGD w/ subsampling | Mini-batch SGD |
|---|---|---|---|---|
| Geometric $\phi$-mixing (Weakly dependent) | $\exp(-k^\theta)$, $\theta > 0$ | $\mathcal{O}(\epsilon^{-2}(\log \epsilon^{-1})^{\frac{2}{\theta}})$ | $\mathcal{O}(\epsilon^{-2}(\log \epsilon^{-1})^{\frac{1}{\theta}})$ | $\mathcal{O}(\epsilon^{-2})$ |
| Fast algebraic $\phi$-mixing (Medium dependent) | $k^{-\theta}$, $\theta \geq 1$ | $\mathcal{O}(\epsilon^{-2-\frac{2}{\theta}})$ | $\mathcal{O}(\epsilon^{-2-\frac{1}{\theta}})$ | $\widetilde{\mathcal{O}}(\epsilon^{-2})$ |
| Slow algebraic $\phi$-mixing (Highly dependent) | $k^{-\theta}$, $0 < \theta < 1$ | $\mathcal{O}(\epsilon^{-2-\frac{2}{\theta}})$ | $\mathcal{O}(\epsilon^{-2-\frac{1}{\theta}})$ | $\mathcal{O}(\epsilon^{-1-\frac{1}{\theta}})$ |

data with geometric ergodicity. Duchi et al. (2011) proved that the stochastic subgradient method has strong convergence guarantee if the mixing time is uniformly bounded. Agarwal & Duchi (2012) studied the convex/strongly convex stochastic optimization problem and proved high-probability convergence bounds for general stochastic algorithms under general stationary mixing processes. Godichon-Baggioni et al. (2021) provided the non-asymptotic analysis of stochastic algorithms with strongly convex objective function over streaming mini-batch data. In a more general setting, the stochastic approximation (SA) problem was studied in (Karimi et al., 2019) by assuming the existence of solution to a Poisson equation. Recently, Debavelaere et al. (2021) developed the asymptotic convergence analysis of SA problem for sub-geometric Markov dynamic noises.

**Finite-time convergence of reinforcement learning**   Recently, a series of work studied the finite-time convergence of many stochastic reinforcement learning algorithms over Markovian dependent samples, including TD learning (Dalal et al., 2018; Xu et al., 2019; Kaledin et al., 2020), Q-learning (Qu & Wierman, 2020; Li et al., 2021; Melo et al., 2008; Chen et al., 2019; Xu & Gu, 2020), fitted Q-iteration (Mnih et al., 2013; 2015; Agarwal et al., 2021), actor-critic algorithms (Wang et al., 2019; Yang et al., 2019; Kumar et al., 2019; Qiu et al., 2019; Wu et al., 2020; Xu et al., 2020), etc. In these studies, the dependent Markovian samples are assumed to satisfy the geometric $\phi$-mixing property, which is satisfied when the underlying Markov chain is uniformly ergodic or time-homogeneous with finite-states.

**Regret of Stochastic Convex Optimization**   There have been many known regret bounds for online convex optimization problem. Hazan (2017) has built the standard $\mathcal{O}(\sqrt{T})$ regret bound for online SGD algorithm with assuming the bounded gradient. Xiao (2009) introduces the regret bound of online dual averaging method. To our best knowledge, there is no high-probability guaranteed regret bound for mini-batch SGD algorithm with considering the data dependence.

## 2   PROBLEM FORMULATION AND ASSUMPTIONS

In this section, we introduce the problem formulation and some basic assumptions. Consider a model with parameters $w$. For any data sample $\xi$, denote $F(w; \xi) \in \mathbb{R}$ as the sample loss of this data sample under the model $w$. In this paper, we consider the following standard stochastic optimization problem that has broad applications in machine learning.

$$\min_{w \in \mathcal{W}} f(w) := \mathbb{E}_{\xi \sim \mu}\big[F(w; \xi)\big]. \tag{P}$$

Here, the expectation is taken over the randomness of the data sample $\xi$, which is drawn from an underlying distribution $\mu$. In particular, we make the following standard assumptions regarding the problem (P) (Agarwal & Duchi, 2012).

**Assumption 2.1.** *The stochastic optimization problem (P) satisfies*

1. *For every $\xi$, function $F(\cdot; \xi)$ is $G$-Lipschitz continuous over $\mathcal{W}$, i.e., for all $w, v \in \mathcal{W}$,*

$$|F(w; \xi) - F(v; \xi)| \leq G\|w - v\|.$$

2. *Function $f(\cdot)$ is convex and bounded below, i.e., $f(w^*) := \inf_{w \in \mathcal{W}} f(w) > -\infty$.*

3. $\mathcal{W}$ is a convex and compact set with bounded diameter $R$, i.e., $\sup_{w,v \in \mathcal{W}} \|w - v\| \leq R$.

To solve this stochastic optimization problem, one often needs to query a set of data samples from the distribution $\mu$ to perform optimization. Unlike traditional stochastic optimization that usually assumes that the data samples are i.i.d. we consider a more general and practical dependent data-generating process as we elaborate below.

**Dependent data-generating process:** We consider a stochastic process $\mathbf{P}$ that generates a stream of data samples $\{\xi_1, \xi_2, ...,\}$, which are not necessarily independent. In particular, the stochastic process $\mathbf{P}$ has an underlying stationary distribution $\mu$. To quantify the dependence of the data generation process, we introduce the following standard $\phi$-mixing model (Agarwal & Duchi, 2012), where we denote $\{\mathcal{F}_t\}_t$ as the canonical filtration generated by $\{\xi_t\}_t$.

**Definition 2.2** ($\phi$-mixing process). Consider a stochastic process $\{\xi_t\}_t$ with a stationary distribution $\mu$. Let $\mathbb{P}(\xi_{t+k} \in \cdot | \mathcal{F}_t)$ be the distribution of the $(t+k)$-th sample conditioned on $\mathcal{F}_t$, and denote $d_{\text{TV}}$ as the total variation distance. Then, the process $\{\xi_t\}_t$ is called $\phi$-mixing if the following mixing coefficient $\phi_\xi(\cdot)$ converges to 0 as $k$ tends to infinity.

$$\phi_\xi(k) := \sup_{t \in \mathbb{N}, A \in \mathcal{F}_t} 2d_{\text{TV}}\big(\mathbb{P}(\xi_{t+k} \in \cdot | A), \mu\big).$$

Intuitively, the $\phi$-mixing coefficient describes how fast the distribution of sample $\xi_{t+k}$ converges to the stationary distribution $\mu$ when conditioned on the filtration $\mathcal{F}_t$, as the time gap $k \to \infty$. The $\phi$-mixing process can be found in many applications, which involve mixing coefficients that converge to zero at different convergence rates. Below we mention some popular examples.

- **Geometric $\phi$-mixing process.** Such a type of process has a geometrically diminishing mixing coefficient, i.e., $\phi_\xi(k) \leq \phi_0 \exp(-ck^\theta)$ for some $\phi_0, c, \theta > 0$. Examples include finite-state ergodic Markov chains and some aperiodic Harris-recurrent Markov processes (Modha & Masry, 1996; Agarwal & Duchi, 2012; Meyn & Tweedie, 2012);

- **Algebraic $\phi$-mixing process.** Such a type of process has a polynomially diminishing mixing coefficient, i.e., $\phi_\xi(k) \leq \phi_0 k^{-\theta}$ for some $\phi_0, \theta > 0$. Examples include a large class of Metropolis-Hastings samplers (Jarner & Roberts, 2002) and some queuing systems (Agarwal & Duchi, 2012).

## 3    CONVERGENCE OF SGD WITH SUBSAMPLING OVER DEPENDENT DATA

In this section, we study the convergence rate and sample complexity of SGD with data subsampling update over $\phi$-mixing dependent data. In Section 3.1, we recap the convergence results of the standard SGD over dependent data established in (Agarwal & Duchi, 2012). In Section 3.2, we establish convergence results of SGD with the data subsampling update.

Throughout, we define the sample complexity as the total number of samples required for the algorithm to output a model $w$ that achieves an $\epsilon$ convergence error, i.e., $f(w) - f(w^*) \leq \epsilon$. Also, the standard regret of a stochastic algorithm is defined as

$$\text{(Regret):} \quad \mathfrak{R}_n := \sum_{t=1}^n F(w(t); \xi_t) - F(w^*; \xi_t),$$

where the models $\{w_1, w_2, ..., w_n\}$ are generated using the data samples $\{\xi_1, \xi_2, ..., \xi_n\}$, respectively, and $w^*$ is the minimizer of $f(w)$. For this sequence of models $\{w_1, w_2, ..., w_n\}$, we make the following mild assumption, which is satisfied by many SGD-type algorithms.

**Assumption 3.1.** *There is a non-increasing sequence $\{\kappa(t)\}_t$ such that $\|w(t+1) - w(t)\| \leq \kappa(t)$.*

### 3.1    STOCHASTIC GRADIENT DESCENT

Stochastic gradient descent (SGD) is a popular and classical algorithm for stochastic optimization. In every iteration $t$, SGD queries a sample $\xi_t$ from the data-generating process and performs the following update.

$$\text{(SGD):} \quad w(t+1) = w(t) - \eta_t \nabla F(w(t); \xi_t), \tag{1}$$

where $\eta_t$ is the learning rate. In Theorem 2 of (Agarwal & Duchi, 2012), the authors established a high probability convergence error bound for a generic class of stochastic algorithms. Specifically, under the Assumptions 2.1 and 3.1, they showed that for any $\tau \in \mathbb{N}$ with probability at least $1 - \delta$, the averaged predictor $\widehat{w}_n := \frac{1}{n} \sum_{t=1}^{n} w(t)$ satisfies

$$f(\widehat{w}_n) - f(w^*) \le \frac{\Re_n}{n} + \frac{(\tau-1)G}{n} \sum_{t=1}^{n} \kappa(t) + \frac{2(\tau-1)GR}{n} + 2GR\sqrt{\frac{2\tau}{n} \log \frac{\tau}{\delta}} + \phi_\xi(\tau)GR. \quad (2)$$

Here, $\Re_n$ is the regret of the algorithm of interest, and $\tau \in \mathbb{N}$ is an auxiliary parameter that is introduced to decouple the dependence of the data samples. From the above bound, one can see that the optimal choice of $\tau$ depends on the convergence rate of the mixing coefficient $\phi_\xi(\tau)$. Specifically, consider the SGD algorithm in (1). It can be shown that it achieves the regret $\Re_n = \mathcal{O}(\sqrt{n})$ and satisfies $\kappa(t) = \mathcal{O}(1/\sqrt{t})$ with a proper diminishing learning rate. Consequently, the above high-probability convergence bound for SGD reduces to

$$f(\widehat{w}_n) - f(w^*) \le \mathcal{O}\Big(\frac{1}{\sqrt{n}} + \inf_{\tau \in \mathbb{N}} \Big\{ \frac{\tau-1}{\sqrt{n}} + \sqrt{\frac{\tau}{n} \log \frac{\tau}{\delta}} + \phi_\xi(\tau) \Big\}\Big). \quad (3)$$

Such a bound further implies the following sample complexity results of SGD under different convergence rates of the mixing coefficient $\phi_\xi$.

**Corollary 3.2.** *The sample complexity of SGD in* (1) *for achieving an $\epsilon$ convergence error over $\phi$-mixing dependent data is given as follows.*

- *If the data is geometric $\phi$-mixing with parameter $\theta > 0$, then we choose $\tau = \mathcal{O}\big((\log \frac{1}{\epsilon})^{\frac{1}{\theta}}\big)$. The resulting sample complexity is in the order of $n = \mathcal{O}\big(\epsilon^{-2}(\log \frac{1}{\epsilon})^{\frac{2}{\theta}}\big)$.*

- *If the data is algebraic $\phi$-mixing with parameter $\theta > 0$, then we choose $\tau = \mathcal{O}(\epsilon^{-\frac{1}{\theta}})$. The resulting sample complexity is in the order of $n = \mathcal{O}(\epsilon^{-2-\frac{2}{\theta}})$.*

It can be seen that if the data-generating process has a fast geometrically diminishing mixing coefficient, i.e., the data samples are close to being independent from each other, then the resulting sample complexity is almost the same as that of SGD with i.i.d. samples. On the other hand, if the data-generating process mixes slowly with an algebraically diminishing mixing coefficient, i.e., the data samples are highly dependent, then the data dependence increases the sample complexity by a non-negligible factor of $\epsilon^{-\frac{2}{\theta}}$. In particular, such a factor is substantially large if the mixing rate parameter $\theta$ is close to zero.

## 3.2 SGD with subsampling

When apply SGD to solve stochastic optimization problems over dependent data, the key challenge is that the data dependence introduces non-negligible bias that slows down the convergence of the algorithm. Hence, a straightforward solution is to reduce data dependence before performing stochastic optimization. In the existing literature, a simple and useful approach is data subsampling (Nagaraj et al., 2020; Kotsalis et al., 2020). Next, we show that such an approach leads to an improved convergence bound and sample complexity of SGD over highly dependent data.

Specifically, consider a stream of $\phi$-mixing data samples $\{\xi_1, \xi_2, \xi_3, \dots\}$. Instead of utilizing the entire stream of data, we subsample a subset of this data stream with period $r \in \mathbb{N}$ and obtain the following subsampled data stream

$$\{\xi_1, \xi_{r+1}, \xi_{2r+1}, \dots\}.$$

In particular, let $\{\mathcal{F}_t\}_t$ be the canonical filtration generated by $\{\xi_{tr+1}\}_t$. Since the consecutive subsampled samples are $r$ time steps away from each other, it is easy to verify that the subsampled data stream $\{\xi_{tr+1}\}_t$ is also a $\phi$-mixing process with mixing coefficient given by $\phi_\xi^r(t) = \phi_\xi(rt)$, where $\phi_\xi^r$ denotes the mixing coefficient of the subsampled data stream $\{\xi_{tr+1}\}_t$. Therefore, by periodically subsampling the data stream, the resulting subsampled process has a faster-converging mixing coefficient. Then, we apply SGD over such subsampled data, i.e.,

$$(\text{SGD with subsampling}): \quad w(t+1) = w(t) - \eta_t \nabla F(w(t); \xi_{tr+1}). \quad (4)$$

In particular, the convergence error bound in eq. (2) still holds by replacing $\phi_\xi(\tau)$ with $\phi_\xi(r\tau)$, and we obtain the following bound for SGD with subsampling.

$$f(\widehat{w}_n) - f(w^*) \leq \mathcal{O}\Big(\frac{1}{\sqrt{n}} + \inf_{\tau\in\mathbb{N}}\Big\{\frac{(\tau-1)}{\sqrt{n}} + \sqrt{\frac{\tau}{n}\log\frac{\tau}{\delta}} + \phi_\xi(r\tau)\Big\}\Big). \tag{5}$$

Such a bound further implies the following sample complexity results of SGD with subsampling under different convergence rates of the mixing coefficient $\phi_\xi$.

**Corollary 3.3.** *The sample complexity of SGD with subsampling in* (4) *for achieving an $\epsilon$ convergence error over $\phi$-mixing dependent data is given as follows.*

- *If the data is geometric $\phi$-mixing with parameter $\theta > 0$, then we choose $r = \mathcal{O}\big((\log\frac{1}{\epsilon})^{\frac{1}{\theta}}\big)$ and $\tau = \mathcal{O}(1)$. The resulting sample complexity is in the order of $rn = \mathcal{O}\big(\epsilon^{-2}(\log\frac{1}{\epsilon})^{\frac{1}{\theta}}\big)$.*

- *If the data is algebraic $\phi$-mixing with parameter $\theta > 0$, then we choose $r = \mathcal{O}\big(\epsilon^{-\frac{1}{\theta}}\big)$ and $\tau = \mathcal{O}(1)$. The resulting sample complexity is in the order of $rn = \mathcal{O}\big(\epsilon^{-2-\frac{1}{\theta}}\big)$.*

Compare the above sample complexity results with those of the standard SGD in Corollary 3.2, we conclude that data-subsampling can improve the sample complexity by a factor of $(\log\frac{1}{\epsilon})^{\frac{1}{\theta}}$ and $\epsilon^{-\frac{1}{\theta}}$ for geometric $\phi$-mixing and algebraic $\phi$-mixing data, respectively. Intuitively, this is because with data subsampling, we can choose a sufficiently large subsampling period $r$ to decouple the data dependence in the term $\phi_\xi(r\tau)$, as opposed to choosing a large $\tau$ in Corollary 3.2. In this way, the order of the dominant term $\sqrt{\frac{\tau}{n}\log\frac{\tau}{\delta}}$ is reduced. Therefore, when the data is highly dependent, it is beneficial to subsample the dependent data before performing SGD. We also note another advantage of using data-subsampling, i.e., it only requires computing the stochastic gradients of the subsampled data, and therefore can substantially reduce the computation load.

## 4 CONVERGENCE OF MINI-BATCH SGD OVER DEPENDENT DATA

Although the data-subsampling update scheme studied in the previous section helps improve the sample complexity of SGD, it does not leverage the full information of all the queried data. In particular, when the data is highly dependent, we need to choose a large period $r$ to reduce data dependence, and this will throw away a huge amount of valuable samples. In this section, we study SGD with another popular update scheme that leverages the full information of all the sampled data, i.e., the mini-batch update scheme. We show that this simple and popular scheme can effectively reduce data dependence without skipping data samples, and can achieve an improved sample complexity over SGD with subsampling.

Specifically, we consider a data stream $\{\xi_t\}_t$ with $\phi$-mixing dependent samples. We rearrange the data samples into a stream of mini-batches $\{x_t\}_t$, where each mini-batch $x_t$ contains $B$ samples, i.e., $x_t = \{\xi_{(t-1)B+1}, \xi_{(t-1)B+2}, \ldots, \xi_{tB}\}$. Then, we perform mini-batch SGD update as follows.

$$\text{(Mini-batch SGD):} \quad w(t+1) = w(t) - \frac{\eta_t}{B}\sum_{\xi\in x_t}\nabla F(w(t);\xi). \tag{6}$$

Performing SGD updates with mini-batch data has several advantages. First, it substantially reduce the optimization variance and allows to use a large learning rate to facilitate the convergence of the algorithm. As a comparison, SGD with subsampling still suffers from a large optimization variance. Second, unlike SGD with subsampling, mini-batch SGD utilizes the information of all the data samples to improve the performance of the model. Moreover, as we show in the following lemma, mini-batch update substantially reduces the stochastic bias caused by the data dependence. In the sequel, we denote $F(w;x) := \frac{1}{B}\sum_{\xi\in x}F(w;\xi)$ as the average loss on a mini-batch of samples. With a bit abuse of notation, we also define $\{\mathcal{F}_t\}_t$ as the canonical filtration generated by the mini-batch samples $\{x_t\}_t$.

**Lemma 4.1.** *Let Assumption 2.1 hold and consider the mini-batch data stream $\{x_t\}_t$. Then, for any $w, v \in \mathcal{W}$ measureable with regard to $\mathcal{F}_t$ and any $\tau \in \mathbb{N}$, it holds that*

$$\mathbb{E}\big[F(w; x_{t+\tau}) - F(v; x_{t+\tau})|\mathcal{F}_t\big] - \big(f(w) - f(v)\big) \leq \frac{GR}{B}\sum_{i=1}^{B}\phi_\xi(\tau B + i). \tag{7}$$

With dependent data, the above lemma shows that we can approximate the population risk $f(w)$ by the conditional expectation $\mathbb{E}[F(w; x_{t+\tau})|\mathcal{F}_t]$, which involves the sample $x_{t+\tau}$ that is $\tau$ steps ahead of the filtration $\mathcal{F}_t$. Intuitively, by the $\phi$-mixing principle, as $\tau$ gets larger, the distribution of $x_{t+\tau}$ conditional on $\mathcal{F}_t$ gets closer to the stationary distribution $\mu$. In general, the estimation bias $\frac{GR}{B} \sum_{i=1}^{B} \phi_\xi(\tau B + i)$ depends on both the batch size and the accumulated mixing coefficient over the corresponding batch of samples. To provide a concrete understanding, below we calculate the estimation bias for several different mixing models.

- **Geometric $\phi$-mixing:** In this case, $\sum_{i=1}^{B} \phi_\xi(\tau B + i) \leq \sum_{i=1}^{\infty} \phi_\xi(i) = \mathcal{O}(1)$. Hence, the estimation bias is in the order of $\mathcal{O}(\frac{GR}{B})$.

- **Fast algebraic $\phi$-mixing ($\theta \geq 1$):** In this case, $\sum_{i=1}^{B} \phi_\xi(\tau B + i) \leq \sum_{i=1}^{\infty} \phi_\xi(i) = \widetilde{\mathcal{O}}(1)$. Hence, the estimation bias is in the order of $\widetilde{\mathcal{O}}(\frac{GR}{B})$, where $\widetilde{\mathcal{O}}$ hides all logarithm factors.

- **Slow algebraic $\phi$-mixing ($0 < \theta < 1$):** In this case, $\sum_{i=1}^{B} \phi_\xi(\tau B + i) \leq \mathcal{O}((\tau B)^{1-\theta})$. Hence, the estimation bias is in the order of $\mathcal{O}(\frac{GR\tau^{1-\theta}}{B^\theta})$.

It can be seen that if the mixing coefficient converges fast, i.e., either geometrically or fast algebraically, then the data dependence has a negligible impact on the estimation error. Consequently, choosing a large batch size can substantially reduce the estimation bias. On the other hand, when the mixing coefficient converges slow algebraically, it substantially increases the estimation bias, but it is still beneficial to use a large batch size. This result shows that mini-batch update can effectively reduce the statistical bias of stochastic approximation for a wide spectrum of dependent data generating processes.

We obtain prove the following convergence error bound for mini-batch SGD over dependent data.

**Theorem 4.2.** *Let Assumption 2.1 and 3.1 hold. Apply mini-batch SGD to solve the stochastic optimization problem (P) over $\phi$-mixing dependent data and assume that it achieves regret $\mathfrak{R}_n$. Then, for any $\tau \in \mathbb{N}$ and any minimizer $w^*$ with probability at least $1 - \delta$, the averaged predictor $\widehat{w}_n := \frac{1}{n} \sum_{t=1}^{n} w(t)$ satisfies*

$$f(\widehat{w}_n) - f(w^*) \leq \frac{\mathfrak{R}_n}{n} + \frac{G(\tau - 1)}{n} \sum_{t=1}^{n-\tau+1} \kappa(t) + \frac{GR(\tau - 1)}{n}$$

$$+ \mathcal{O}\left( \frac{1}{nB} \sum_{i=1}^{B} \phi(\tau B + i) + \sqrt{\frac{\tau}{nB} \log \frac{\tau}{\delta}} \log \frac{n}{\delta} \left( B^{-\frac{1}{4}} + \left[ \sum_{i=1}^{B} \phi(i) \right]^{\frac{1}{4}} \right) \right). \quad (8)$$

To further understand the order of the above bound, a standard regret analysis shows that mini-batch SGD achieves a regret in the order of $\frac{\mathfrak{R}_n}{n} = \widetilde{\mathcal{O}}(\sqrt{\frac{\sum_{j=1}^{n} \phi(j)}{nB}})$ and $\kappa(t) \equiv \mathcal{O}(\sqrt{\frac{B}{n}})$ (see Theorem C.3 for the proof). Consequently, the above convergence error bound reduces to the following bound, where we hide all logarithm factors for simplicity of presentation.

$$f(\widehat{w}_n) - f(w^*) \leq \widetilde{\mathcal{O}}\left( \sqrt{\frac{\sum_{j=1}^{n} \phi(j)}{nB}} + \frac{GR(\tau - 1)}{n} \right. \qquad (9)$$

$$\left. + \frac{1}{nB} \sum_{i=1}^{B} \phi(\tau B + i) + \sqrt{\frac{\tau}{nB}} \left( B^{-\frac{1}{4}} + \left[ \sum_{i=1}^{B} \phi(i) \right]^{\frac{1}{4}} \right) \right). \qquad (10)$$

Such a bound further implies the following sample complexity results of mini-batch SGD under different convergence rates of the mixing coefficient $\phi_\xi$.

**Corollary 4.3.** *The sample complexity of mini-batch SGD in (6) for achieving an $\epsilon$ convergence error over $\phi$-mixing dependent data is given as follows.*

- *If the data is geometric $\phi$-mixing with parameter $\theta > 0$, then we choose $\tau = 1, B = \mathcal{O}(\epsilon^{-1}), n = \mathcal{O}(\epsilon^{-1})$. The overall sample complexity is $nB = \mathcal{O}(\epsilon^{-2})$.*

- *If the data is fast algebraic $\phi$-mixing with parameter $\theta \geq 1$, then we choose $\tau = 1, B = \mathcal{O}(\epsilon^{-1}), n = \mathcal{O}(\epsilon^{-1})$. The overall sample complexity is $nB = \widetilde{\mathcal{O}}(\epsilon^{-2})$.*

- *If the data is slow algebraic $\phi$-mixing with parameter $0 < \theta < 1$, then we choose $\tau = 1, B = \mathcal{O}(\epsilon^{-\frac{1}{\theta}}), n = \mathcal{O}(\epsilon^{-1})$. The overall sample complexity is $nB = \mathcal{O}(\epsilon^{-1-\frac{1}{\theta}})$.*

It can be seen that mini-batch SGD can achieve an order-wise lower sample complexity than the SGD with subsampling in the full spectrum of $\phi$-mixing convergence rate. Specifically, mini-batch SGD improves the sample complexity over that of SGD with subsampling by a factor of $\mathcal{O}((\log \frac{1}{\epsilon})^{\frac{1}{\theta}})$, $\widetilde{\mathcal{O}}(\epsilon^{-\frac{1}{\theta}})$ and $\mathcal{O}(\epsilon^{-1})$ for geometric $\phi$-mixing, fast algebraic $\phi$-mixing and slow algebraic $\phi$-mixing data samples, respectively. This shows that mini-batch update can effectively reduce the bias caused by data dependence and leverage the full information of all the data samples to improve the learning performance.

To intuitively explain, this is because with mini-batch updates, we can choose a sufficiently large batch size $B$ to reduce the bias caused by the data dependence and choose a small auxiliary parameter $\tau = 1$. As a comparison, to control the bias caused by data dependence, the standard SGD needs to choose a very large $\tau$ and the SGD with subsampling needs to choose a large subsampling period $r$ that skips a huge amount of valuable data samples, especially when the mixing coefficient converges slowly. Therefore, our result proves that it is beneficial to use mini-batch stochastic gradient updates when the data samples are highly dependent.

We note that our proof of the tight high-probability bound in Theorem 4.2 for mini-batch SGD involves substantial new developments compared with the proof of (Agarwal & Duchi, 2012). Next, we elaborate on our technical novelty.

- In (Agarwal & Duchi, 2012), they defined the following random variable

$$X_t^i := f\big(w((t-1)\tau + i)\big) - f(w^*) + F\big(w((t-1)\tau + i); \xi_{t+\tau-1}\big) - F\big(w^*; \xi_{t+\tau-1}\big).$$

As this random variable involves only one sample $\xi_{t+\tau-1}$, they bound the bias term $X_t^i - \mathbb{E}[X_t^i | \mathcal{F}_{t-1}^i]$ as a universal constant. As a comparison, the random variable $X_t^i$ would involve a mini-batch of samples $x_{t+\tau-1}$ in our analysis. With the mini-batch structure, the bias $X_t^i - \mathbb{E}[X_t^i | \mathcal{F}_{t-1}^i]$ can be written as an average of $B$ zero-mean dependent random variables, which is close to zero with high probability due to the concentration phenomenon. Consequently, we are able to apply a Bernstein-type inequality developed in (Delyon et al., 2009) for dependent stochastic process to obtain an improved bias bound from $\mathcal{O}(1)$ to $\widetilde{\mathcal{O}}(1/\sqrt{B})$. This is critical for obtaining the improved bound.

- Second, with the improved high-probability bias bound mentioned above, the remaining proof of (Agarwal & Duchi, 2012) no longer holds. Specifically, we can no longer apply the Azuma's inequality to bound the accumulated bias $\sum_t (X_t^i - \mathbb{E}[X_t^i | \mathcal{F}_{t-1}^i])$, as each bias term is no longer bounded with probability one. To address this issue, we developed a generalized Azuma's inequality for martingale differences in Lemma B.3 based on Proposition 34 of (Tao et al., 2015) for independent zero-mean random variables.

- Third, we develop a high-probability regret bound for mini-batch SGD over dependent data so that it can be integrated with the high-probability convergence bound in Theorem 4.2. To our best knowledge, the regret of SGD over dependent data has not been studied before.

## 5   NUMERICAL EXAMPLE

We examine our theory via a basic convex quadratic optimization problem, which is written as

$$\min_{w \in \mathbb{R}^d} f(w) := \mathbb{E}_{\xi \sim \mu}\big[(w - \xi)^\top A(w - \xi)\big],$$

where $A \succeq 0$ is a fixed positive semi-definite matrix and $\mu$ is the uniform distribution on $[0,1]^d$. Then, following the construction in (Jarner & Roberts, 2002), we generate an algebraic $\phi$-mixing Markov chain that has the stationary distribution $\mu$. In particular, its mixing coefficient $\phi_\xi(k)$ converges at a sublinear convergence rate $k^{-\frac{1}{r}}$, where $r > 0$ is a parameter that controls the speed of convergence. Please refer to Appendix D for more details of the experiment setup.

We first estimate the following stochastic bias at the fixed origin point $w = \mathbf{0}_d$.

$$(\text{Bias}): \quad \Big|\mathbb{E}\big[F(w; x_\tau) | x_0 = \mathbf{0}_d\big] - f(w)\Big|,$$

where the expectation is taken over the randomness of the mini-batch of samples queried at time $\tau \in \mathbb{N}$. Such a bias is affected by several factors, including the time gap $\tau$, the batch size $B$ and the convergence rate parameter $r$ of the mixing coefficient.

In Figure 1, we investigate the impact of these factors on the stochastic bias, and we use 10k Monte Carlo samples to estimate the stochastic bias. The left two figures fix the batch size, and it can be seen that the bias decreases as $\tau$ increases, which matches the definition of the $\phi$-mixing property. Also, a faster-mixing Markov chain (i.e., smaller $r$) leads to a smaller bias. In particular, with batch size $B = 1$ and a slow-mixing chain $r = 2$, it takes an unacceptably large $\tau$ to achieve a relatively small bias. This provides an empirical justification to Corollary 3.2 and explains why the standard SGD suffers from a high sample complexity over highly dependent data. Moreover, as the batch size gets larger, one can achieve a numerically smaller bias, which matches our Lemma 4.1. The right two figures fix the convergence rate parameter of the mixing coefficient, and it can be seen that increasing the batch size significantly reduces the bias. Consequently, instead of choosing a large $\tau$ to reduce the bias, one can simply choose a large batch size $B = 100$ and set $\tau = 1$. This observation matches and justifies our theoretical results in Corollary 4.3.

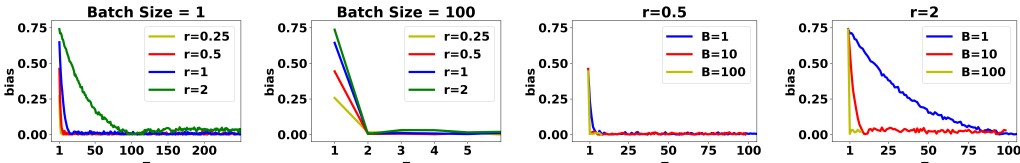

Figure 1: Impact of $\tau$, batch size $B$ and convergence rate of mixing coefficient on the bias.

We further compare the convergence of SGD, SGD with subsampling and mini-batch SGD. Here, we set $r = 2$ to generate highly dependent data samples. We set learning rate $\eta = 0.01$ for both SGD and SGD with subsampling, and set learning rate $\eta = 0.01 \times \sqrt{\frac{B}{\sum_{j=1}^{B} \phi_\xi(j)}} = 0.01 \times 100^{1/4}$ for mini-batch SGD with batch size $B = 100$, as suggested by Theorem C.3 in the appendix. The results are plotted in Figure 2, where each curve corresponds to the mean of 100 independent trails. It can be seen that SGD with subsampling achieves a lower loss than the standard SGD asymptotically, due to the use of less dependent data. Moreover, mini-batch SGD achieves the smallest asymptotic loss. All these observations are consistent with our theoretical results.

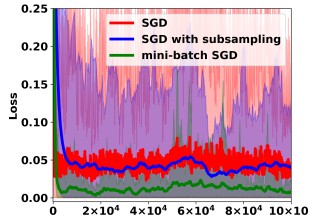

Figure 2: Comparison of sample complexity of different SGD algorithms.

# 6 CONCLUSION

In this study, we investigate the convergence property of SGD under various popular stochastic update schemes over highly dependent data. Unlike the conventional i.i.d. data setting in which the stochastic update schemes do not affect the sample complexity of SGD, the convergence of SGD in the data-dependent setting critically depends on the structure of the stochastic update scheme. In particular, we show that both data subsampling and mini-batch sampling can substantially improve the sample complexity of SGD over highly dependent data. Our study takes one step forward toward understanding the theoretical limits of stochastic optimization over dependent data, and it opens many directions for future study. For example, it is interesting to further explore the impact of algorithm structure on the sample complexity of stochastic reinforcement learning algorithms. Also, it is important to develop advanced algorithm update schemes that can facilitate the convergence of learning over highly dependent data.

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

# Appendix

## Table of Contents

**Notation:** To simplify the notation, throughout the appendix, we denote $\xi_t^{(i)} := \xi_{(t-1)B+i}$, which corresponds to the $i$-th data sample of the $t$-th mini-batch data $x_t$. With this notation, we have $x_t = \{\xi_t^{(1)}, \xi_t^{(2)}, ..., \xi_t^{(B)}\}$.

## A   PROOF OF COROLLARY 3.3

In this section, we analyze the convergence error bound of the SGD with data-subsampling in (4).

Given a $\phi_\xi$-mixing data stream $\{\xi_1, \xi_2, \xi_3, \dots\}$, we consider the following subsampled data stream
$$\{\xi_1, \xi_{r+1}, \xi_{2r+1}, \dots\}.$$
Let $\mathcal{F}$ be the canonical filtration generated by $\{x_t\}$. Then the subsampled data stream $\{\xi_{tr+1}\}_t$ is $\phi_\xi^r$-mixing with the mixing coefficient given by
$$\phi_\xi^r(t) = \phi_\xi(rt).$$
With this mixing coefficient, we can apply Theorem 2 of (Agarwal & Duchi, 2012) and obtain the following convergence error bound for any $\tau \in \mathbb{N}$.

$$f(\widehat{w}_n) - f(w^*) \leq \mathcal{O}\Big( \frac{\mathfrak{R}_n}{n} + \frac{(\tau-1)}{n} \sum_{t=1}^n \kappa(t) + \frac{\tau}{n} + \sqrt{\frac{\tau}{n} \log \frac{\tau}{\delta}} + \phi_\xi(r\tau) \Big).$$

Consider the standard SGD with a diminishing learning rate, we have $\kappa(t) = \mathcal{O}(\frac{1}{\sqrt{t}})$ and $\mathfrak{R}_n = \mathcal{O}(\sqrt{n})$. Then, the convergence error bound becomes

$$f(\widehat{w}_n) - f(w^*) \leq \mathcal{O}\Big( \frac{1}{\sqrt{n}} + \inf_{\tau \in \mathbb{N}} \Big\{ \frac{(\tau-1)}{\sqrt{n}} + \sqrt{\frac{\tau}{n} \log \frac{\tau}{\delta}} + \phi_\xi(r\tau) \Big\} \Big).$$

The above result further implies the following sample complexity results for different convergence rates of the mixing coefficient.

- **Geometric $\phi$-mixing:** In this case, $\phi_\xi(k) \leq \mathcal{O}(\exp(-k^\theta))$ for some $\theta > 0$. Set the last term $\phi_\xi(r\tau) = \mathcal{O}(\epsilon)$. We obtain that $r\tau = \mathcal{O}((\log \frac{1}{\epsilon})^{\frac{1}{\theta}})$. Further set the second term $\frac{\tau-1}{\sqrt{n}} = \mathcal{O}(\epsilon)$. We obtain that $n\tau^{-2} = \mathcal{O}(\epsilon^{-2})$. By choosing $\tau = \mathcal{O}(1)$, the sample complexity is in the order of

$$\epsilon\text{-complexity} = r \cdot n = \mathcal{O}\Big( \Big( \log \frac{1}{\epsilon} \Big)^{\frac{1}{\theta}} \tau^2 \epsilon^{-2} \Big) = \mathcal{O}\Big( \epsilon^{-2} \Big( \log \epsilon^{-1} \Big)^{\frac{1}{\theta}} \Big).$$

- **Algebraic $\phi$-mixing:** In this case, $\phi_\xi(k) \leq \mathcal{O}(k^{-\theta})$ for some $\theta > 0$. Set the last term $\phi_\xi(r\tau) = \mathcal{O}(\epsilon)$. We obtain that $\tau r = \mathcal{O}(\epsilon^{-\frac{1}{\theta}})$. Set the second term $\frac{\tau-1}{\sqrt{n}} = \mathcal{O}(\epsilon)$. We obtain that $n\tau^{-2} = \mathcal{O}(\epsilon^{-2})$. By setting $\tau = \mathcal{O}(1)$, the sample complexity is in the order of

$$\epsilon\text{-complexity} = r \cdot n = \mathcal{O}(\epsilon^{-\frac{1}{\theta}} \tau^2 \epsilon^{-2}) = \mathcal{O}(\epsilon^{-2-\frac{1}{\theta}}).$$

# B    PROOF OF THEOREM 4.2

## B.1    KEY LEMMAS

In this subsection, we present several useful preliminary results for proving Theorem 4.2. Define $\mathbb{N} := \{1, 2, 3, \dots\}$. Throughout this subsection, we assume that Assumption 2.1 holds. The following lemma is a generalization of the Lemma 1 in (Agarwal & Duchi, 2012).

**Lemma B.1.** *Let $w, v$ be measurable with respect to $\mathcal{F}_t$. Then for any $\tau \in \mathbb{N}$,*

$$\mathbb{E}\big[F(w; x_{t+\tau}) - F(v; x_{t+\tau})|\mathcal{F}_t\big] \leq \frac{GR}{B} \sum_{i=1}^{B} \phi_\xi(\tau B + i) + f(w) - f(v).$$

*Proof.* For any $\tau \in \mathbb{N}$, we consider the following decomposition.

$$\begin{aligned}
&\mathbb{E}[F(w; x_{t+\tau}) - F(v; x_{t+\tau})|\mathcal{F}_t]\\
=&\mathbb{E}[F(w; x_{t+\tau}) - f(w) + f(v) - F(v; x_{t+\tau})|\mathcal{F}_t] + f(w) - f(v)\\
=&\underbrace{\Big[\frac{1}{B}\sum_{i=1}^{B}\int F(w; \xi_{t+\tau}^{(i)})\mathrm{d}\mathbb{P}(\xi_{t+\tau}^{(i)} \in \cdot|\mathcal{F}_t) - \int F(w; \xi)\mathrm{d}\mu\Big] - \Big[\frac{1}{B}\sum_{i=1}^{B}\int F(v; \xi_{t+\tau}^{(i)})\mathrm{d}\mathbb{P}(\xi_{t+\tau}^{(i)} \in \cdot|\mathcal{F}_t) - \int F(v; x)\mathrm{d}\mu\Big]}_{(A)}\\
&+ f(w) - f(v).
\end{aligned}$$

We can further bound the term (A) using the mixing property of the dependent data stream.

$$\begin{aligned}
(A) =& \Big[\frac{1}{B}\sum_{i=1}^{B}\int F(w; \xi_{t+\tau}^{(i)})\mathrm{d}\mathbb{P}(\xi_{t+\tau}^{(i)} \in \cdot|\mathcal{F}_t) - \int F(w; \xi)\mathrm{d}\mu\Big] - \Big[\frac{1}{B}\sum_{i=1}^{B}\int F(v; \xi_{t+\tau}^{(i)})\mathrm{d}\mathbb{P}(\xi_{t+\tau}^{(i)} \in \cdot|\mathcal{F}_t) - \int F(v; x)\mathrm{d}\mu\Big]\\
=& \frac{1}{B}\sum_{i=1}^{B}\int (F(w; \xi) - F(v; \xi))\mathrm{d}\big(\mathbb{P}(\xi_{t+\tau}^{(i)} \in \mathrm{d}\xi|\mathcal{F}_t) - \mu(\mathrm{d}\xi)\big)\\
\leq& \frac{1}{B}\sum_{i=1}^{B}\int GR\mathrm{d}\big|\mathbb{P}(\xi_{t+\tau}^{(i)} \in \mathrm{d}\xi|\mathcal{F}_t) - \mu(\mathrm{d}\xi)\big|\\
\leq& \frac{GR}{B}\sum_{i=1}^{B}\phi_\xi(\tau B + i),
\end{aligned}$$

where in the first inequality we use the facts that $F(\cdot; \xi)$ is $G$-Lipschitz and the domain is bounded by $R$, and the second inequality is implied by the $\phi$-mixing property. Substituting the above upper bound of (A) into the previous equation yields that

$$\mathbb{E}[F(w; x_{t+\tau}) - F(v; x_{t+\tau})|\mathcal{F}_t] \leq \frac{GR}{B}\sum_{i=1}^{B}\phi_\xi(\tau B + i) + f(w) - f(v).$$

This completes the proof.    □

**Proposition B.2.** *Let $\{w(t)\}_{t\in\mathbb{N}}$ be the model parameter sequence generated by (6). Also suppose that Assumption 3.1 holds. Then for any $\tau \in \mathbb{N}$, we have*

$$\sum_{t=1}^{n}[f(w(t)) - f(w^*)]$$

$$\leq \sum_{t=1}^{n}[f(w(t)) - F(w(t); x_{t+\tau-1}) + F(w^*; x_{t+\tau-1}) - f(w^*)] + \mathfrak{R}_n + G(\tau - 1)\sum_{t=1}^{n-\tau+1}\kappa(t) + GR(\tau - 1).$$

*Proof.* For any $\tau \in \mathbb{N}$, we consider the following decomposition,

$$\sum_{t=1}^{n}[f(w(t)) - f(w^*)]$$

$$= \sum_{t=1}^{n}[f(w(t)) - F(w(t); x_{t+\tau-1}) + F(w^*; x_{t+\tau-1}) - f(w^*) + F(w(t); x_{t+\tau-1}) - F(w^*; x_{t+\tau-1})]$$

$$= \sum_{t=1}^{n}[f(w(t)) - F(w(t); x_{t+\tau-1}) + F(w^*; x_{t+\tau-1}) - f(w^*)] \quad (11)$$

$$+ \underbrace{\sum_{t=1}^{n}\left[F(w(t); x_{t+\tau-1}) - F(w^*; x_{t+\tau-1})\right]}_{(B)}.$$

We will keep the first term and bound the term (B).

$$(B) = \sum_{t=1}^{n} F(w(t); x_{t+\tau-1}) - F(w^*; x_{t+\tau-1})$$

$$= \underbrace{\sum_{t=1}^{n}[F(w(t); x_t) - F(w^*; x_t)]}_{(B_1)} + \underbrace{\sum_{t=1}^{n-\tau+1}[F(w(t); x_{t+\tau-1}) - F(w(t+\tau-1); x_{t+\tau-1})]}_{(B_2)}$$

$$+ \underbrace{\sum_{t=n-\tau+2}^{n} F(w(t); x_{t+\tau-1}) - \sum_{t=1}^{\tau-1} F(w(t); x_t) + \sum_{t=1}^{\tau-1} F(w^*; x_t) - \sum_{t=n+1}^{n+\tau-1} F(w^*; x_t)}_{(B_3)}.$$

Recall that the term $(B_1)$ is the regret $\mathfrak{R}_n$. We can bound the term $(B_2)$ by noting that

$$F(w(t); x_{t+\tau-1}) - F(w(t+\tau-1); x_{t+\tau-1}) \le G\|w(t+\tau-1) - w(t)\|$$

$$\le G\sum_{i=0}^{\tau-2}\|w(t+i+1) - w(t+i)\|$$

$$\le G\sum_{i=0}^{\tau-2}\kappa(t+i)$$

$$\le G(\tau-1)\kappa(t).$$

For the term $(B_3)$, we can bound it using the $G$-Lipschitzness of $F(\cdot; \xi)$ and the $R$-bounded domain.

$$\sum_{t=n-\tau+2}^{n} F(w(t); x_{t+\tau-1}) - \sum_{t=1}^{\tau-1} F(w(t); x_t) + \sum_{t=1}^{\tau-1} F(w^*; x_t) - \sum_{t=n+1}^{n+\tau-1} F(w^*; x_{t+\tau-1})$$

$$= \left[\sum_{t=n-\tau+2}^{n} F(w(t); x_{t+\tau-1}) - \sum_{t=n+1}^{n+\tau-1} F(w^*; x_{t+\tau-1})\right] - \left[\sum_{t=1}^{\tau-1} F(w(t); x_t) - \sum_{t=1}^{\tau-1} F(w^*; x_t)\right]$$

$$\le G\left[\sum_{t=n-\tau+2}^{n}\|w(t) - w^*\|\right] + G\left[\sum_{t=1}^{\tau-1}\|w(t) - w^*\|\right]$$

$$\le GR(\tau-1).$$

Combining the above bounds of $(B_1)$, $(B_2)$, and $(B_3)$, we obtain the upper bound of $(B)$ as follows.

$$\sum_{t=1}^{n} F(w(t); x_{t+\tau-1}) - F(w^*; x_{t+\tau-1})$$

$$= \underbrace{\sum_{t=1}^{n} [F(w(t); x_t) - F(w^*; x_t)]}_{(B_1)} + \underbrace{\sum_{t=1}^{n-\tau+1} [F(w(t); x_{t+\tau-1}) - F(w(t+\tau-1); x_{t+\tau-1})]}_{(B_2)}$$

$$+ \underbrace{\sum_{t=n-\tau+2}^{n} F(w(t); x_{t+\tau-1}) - \sum_{t=1}^{\tau-1} F(w(t); x_t) + \sum_{t=1}^{\tau} F(w^*; x_t) - \sum_{t=n+1}^{n+\tau-1} F(w^*; x_t)}_{(B_3)}.$$

$$\leq R_n + G(\tau-1) \sum_{t=1}^{n-\tau+1} \kappa(t) + GR(\tau-1).$$

Then the proof is completed by substituting the upper bound of $(B)$ into (11). $\qquad\square$

The following generalized Azuma's inequality generalizes the Proposition 34 of (Tao et al., 2015). The inequality can be used to bound sum of martingale difference random variables.

**Lemma B.3** (Generalized Azuma's Inequality). *Let $\{X_t\}$ be a martingale difference sequence with respect to its canonical filtration $\mathcal{F}$. Define $Y = \sum_{i=1}^{T} X_i$ and assume $\mathbb{E}|Y| < \infty$. Then for any $\{\alpha_t\}_t > 0$,*

$$\mathbb{P}\left(|Y - \mathbb{E}Y| \geq \lambda \sqrt{\sum_{t=1}^{T} \alpha_t^2}\right) \leq 2 \exp\left(-\frac{\lambda^2}{2}\right) + \sum_{t=1}^{T} \mathbb{P}(|X_t| \geq \alpha_t).$$

*Proof.* Let $\mathcal{T} := \min\{t : |X_t| > \alpha_t\}$. Then the sets $B_t := \{\omega : \mathcal{T}(\omega) = t\}$ are disjoint. Construct

$$Y'(\omega) := \begin{cases} Y(\omega) & \text{if } \omega \in \left(\bigcup_{t=1}^{T} B_t\right)^C, \\ \mathbb{E}[Y|B_t] & \text{if } \omega \in B_t \text{ for all } t \in \{1, 2, \ldots, T\}. \end{cases}$$

By the above construction, the associated Doob martingale of $Y'$ with respect to $\mathcal{F}$ is $\{Z_t := \sum_{i=1}^{t \wedge \mathcal{T}} X_i\}$. It satisfies the conditions of Azuma's inequality, i.e.,

- $\{Z_t\}$ forms a martingale with respect to $\mathcal{F}$ (because the stopped martingale is still a martingale).

- $|Z_t - Z_{t-1}| \leq \alpha_t$.

Then we can apply Azuma's inequality to $Y'$.

$$\mathbb{P}\left(|Y' - \mathbb{E}Y'| \geq \lambda \sqrt{\sum_{t=1}^{T} \alpha_t^2}\right) \leq 2 \exp\left(-\frac{\lambda^2}{2}\right).$$

Now we can bound $\mathbb{P}\left(|Y - \mathbb{E}Y| \geq \lambda\sqrt{\sum_{t=1}^{T} \alpha_t^2}\right)$ as follows.

$$
\mathbb{P}\left(|Y - \mathbb{E}Y| \geq \lambda\sqrt{\sum_{t=1}^{T} \alpha_t^2}\right)
$$

$$
= \mathbb{P}\left(|Y - \mathbb{E}Y| \geq \lambda\sqrt{\sum_{t=1}^{T} \alpha_t^2}, \, Y = Y'\right) + \mathbb{P}\left(|Y - \mathbb{E}Y| \geq \lambda\sqrt{\sum_{t=1}^{T} \alpha_t^2}, \, Y \neq Y'\right)
$$

$$
\leq \mathbb{P}\left(|Y' - \mathbb{E}Y'| \geq \lambda\sqrt{\sum_{t=1}^{T} \alpha_t^2}\right) + \mathbb{P}\left(Y \neq Y'\right)
$$

$$
\leq 2\exp\left(-\frac{\lambda^2}{2}\right) + \sum_{t=1}^{T} \mathbb{P}(|X_t| \geq \alpha_t).
$$

Then the proof is completed. Here we notice the fact that $\mathbb{E}Y' = \mathbb{E}Y$ by our construction. $\quad\square$

The following lemma is taken from (22), Theorem 4 of (Delyon et al., 2009).

**Lemma B.4** (Bernstein's Inequality for Dependent Process)**.** *Let $\{Z_t\}$ be a centered adaptive process with respect to $\mathcal{F}$. Define the following quantities.*

$$
q = \sum_{k=1}^{n} \sum_{i=1}^{k-1} \|Z_i\|_{\infty} \cdot \|\mathbb{E}[Z_k|\mathcal{F}_i]\|_{\infty},
$$

$$
v = \sum_{k} \|\mathbb{E}[Z_k^2|Z_{k-1}, \ldots, Z_1]\|_{\infty},
$$

$$
m = \sup_{1 \leq i \leq n} \|Z_i\|_{\infty}.
$$

*Then, it holds that*

$$
\mathbb{P}\Big(\sum_{i=1}^{n} Z_i \geq t\Big) \leq \exp\left(-\frac{t^2}{2(v+2q) + 2tm/3}\right).
$$

**Application of Lemma B.4 to our proof.** Here we make some comments about how to apply this inequality in our main proof. We define the following random variable in our proof. Throughout, we use the batch-level filtration $\mathcal{F}$ and the intra-batch level filtration $\widehat{\mathcal{F}}$. The formal definition is given in Section B.2.

$$
X_t^i = f\big(w((t-1)\tau + 1)\big) - f(w^*) + F(w^*; x_{t\tau+i-1}) - F\big(w((t-1)\tau + 1); x_{t\tau+i-1}\big).
$$

We also define the filtration $\mathcal{F}_t^i := \mathcal{F}_{t\tau+i-1}$ for simplicity. Then, we have

$$
\mathbb{E}[X_t^i|\mathcal{F}_{t-1}^i] = f\big(w((t-1)\tau + 1)\big) - f(w^*) + \mathbb{E}\big[F(w^*; x_{t\tau+i-1}) - F\big(w((t-1)\tau + 1); x_{t\tau+i-1}\big)|\mathcal{F}_{t-1}^i\big].
$$

Then, the bias can be rewritten as

$$
\begin{aligned}
&X_t^i - \mathbb{E}[X_t^i|\mathcal{F}_{t-1}^i] \\
&= F(w^*; x_{t\tau+i-1}) - F(w((t-1)\tau + 1); x_{t\tau+i-1}) - \mathbb{E}\big[F(w^*; x_{t\tau+i-1}) - F\big(w((t-1)\tau + 1); x_{t\tau+i-1}\big)|\mathcal{F}_{t-1}^i\big] \\
&= \frac{1}{B} \sum_{\xi \in x_{t\tau+i-1}} Y_t^i(\xi),
\end{aligned}
$$

where $Y_t^i$ is defined as

$$
Y_t^i(\xi) = F(w^*; \xi) - F\big(w((t-1)\tau + 1); \xi\big) - \mathbb{E}\big[F(w^*; \xi) - F\big(w((t-1)\tau + 1); \xi\big)|\mathcal{F}_{t-1}^i\big].
$$

More specifically, we have

$$X_t^i - \mathbb{E}[X_t^i | \mathcal{F}_{t-1}^i] = \frac{1}{B} \sum_{\xi \in x_{t\tau+i-1}} Y_t^i(\xi) = \frac{1}{B} \sum_{j=1}^{B} Y_t^i(\xi_{t\tau+i-1}^{(j)}).$$

Recall that $\widehat{\mathcal{F}}$ is the canonical filtration generated from the data stream (12). Moreover, $\{Y_t^i(\xi_{t\tau+i-1}^{(j)})\}_{j=1,2,\dots,B}$ is centered and adaptive with respect to this filtration. Then we can evaluate the quantities $q$, $v$, and $m$ in Lemma B.4 as follows.

- Bounding $m$ is simple. By Assumption 2.1 we have $\|Y_t^i(\xi_{t\tau+i-1}^{(j)})\| \le 2GR$.

- The above bound of $m$ leads to a simple bound for $v$, i.e., $v \le 2nG^2R^2$.

- The quantity $q$ can be bounded as follows.

$$
\begin{aligned}
q &:= \sum_{k=1}^{n} \sum_{j=1}^{k-1} \|Y_t^i(\xi_{t\tau+i-1}^{(j)})\|_\infty \|\mathbb{E}[Y_t^i(\xi_{t\tau+i-1}^{(k)})|\widehat{\mathcal{F}}_{t\tau+i-1}^{(j)}]\|_\infty \\
&\le 2GR \sum_{k=1}^{n} \sum_{j=1}^{k-1} \|\mathbb{E}[Y_t^i(\xi_{t\tau+i-1}^{(k)})|\widehat{\mathcal{F}}_{t\tau+i-1}^{(j)}]\|_\infty \\
&= 2GR \sum_{k=1}^{n} \sum_{j=1}^{k-1} \|\mathbb{E}[Y_t^i(\xi_{t\tau+i-1}^{(k)})|\widehat{\mathcal{F}}_{t\tau+i-1}^{(j)}] - \mathbb{E}_{\xi \sim \mu} Y_t^i(\xi_{t\tau+i-1}^{(k)})\|_\infty \\
&\le 4G^2R^2 \sum_{k=1}^{n} \sum_{i=1}^{k-1} \phi_\xi(k-i) \\
&\le 4G^2R^2 n \sum_{i=1}^{n} \phi_\xi(i).
\end{aligned}
$$

Then, by applying Lemma B.4, we obtain the following high-probability bound.

$$
\begin{aligned}
\mathbb{P}\left(|X_t^i - \mathbb{E}[X_t^i|\mathcal{F}_{t-1}^i]| \ge t\right) &\le 2\exp\left(-\frac{B^2t^2}{2(v+2q)+2Btm/3}\right) \\
&\le 2\exp\left(-\frac{B^2t^2}{2(2G^2R^2B + 8G^2R^2B\sum_{i=1}^{B}\phi_\xi(i)) + 4GRBt/3}\right) \\
&= 2\exp\left(-\frac{Bt^2}{2(2G^2R^2 + 8G^2R^2\sum_{i=1}^{B}\phi_\xi(i)) + 4GRt/3}\right).
\end{aligned}
$$

Simplifying yields that

$$\mathbb{P}\left(|X_t^i - \mathbb{E}[X_t^i|\mathcal{F}_{t-1}^i]| \ge t\right) \le 2\exp\left(-\frac{Bt^2}{C + \frac{4}{3}GRt + 16G^2R^2\sum_{i=1}^{B}\phi_\xi(i)}\right),$$

where $C := 4G^2R^2$.

## B.2 PROOF OF THE MAIN RESULT

Recall that we are considering a data stream divided into small mini-batches. For convenience, we re-label the data stream $\{\xi_1, \xi_2, \xi_3, \dots\}$ as follows to explicitly indicate its mini-batch index.

$$\{\xi_1^{(1)}, \xi_1^{(2)}, \dots, \xi_1^{(B)}, \xi_2^{(1)}, \xi_2^{(2)}, \dots, \xi_2^{(B)}, \dots\}. \tag{12}$$

The canonical filtration generated by the re-labeled data stream is denoted by $\widehat{\mathcal{F}}$. Also, when the batch size is clear in the context, we denote the data in the specified mini-batch as $x$. For example,

we use $x_t$ to represent the $t$-th mini-batch $\{\xi_t^{(1)}, \xi_t^{(2)}, \ldots, \xi_t^{(B)}\}$. Then we can re-writhe the above data stream as

$$\{x_1, x_2, x_3, \ldots\}.$$

We denote the canonical filtration generated by the above sequence as $\mathcal{F}$. Note that we have the following relation:

$$\mathcal{F}_t = \widehat{\mathcal{F}}_t^{(B)}.$$

In summary, when we analyze the mini-batch SGD dynamics, we use the filtration $\mathcal{F}$, and when we need to consider intra-batch samples, we use the filtration $\widehat{\mathcal{F}}$.

**Theorem B.5.** *Let $\{w(t)\}_{t \in \mathbb{N}}$ be the model parameter sequence generated by (6). Suppose Assumptions 2.1 and 3.1 hold. Then, for any $\tau \in \mathbb{N}$, with probability at least $1 - \delta$, we have*

$$\sum_{t=1}^{n}[f(w(t)) - f(w^*)]$$

$$\leq GR\frac{n}{B}\sum_{i=1}^{B}\phi_\xi(\tau B + i)$$

$$+ \sqrt{\frac{2\tau n}{B} \cdot \left(\frac{2}{3}\frac{GR}{B}\log\frac{4n}{\delta} + \sqrt{\frac{4}{9}\frac{G^2R^2}{B}(\log\frac{4n}{\delta})^2 + \left(4G^2R^2 + 16G^2R^2\sum_{i=1}^{B}\phi_\xi(i)\right)\log\frac{4n}{\delta}}\right) \cdot \log\frac{4\tau}{\delta}\log\frac{4n}{\delta}}$$

$$+ \mathfrak{R}_n + G(\tau - 1)\sum_{t=1}^{n-\tau+1}\kappa(t) + GR(\tau - 1).$$

*In particular, if $\tau = 1$, then*

$$\sum_{t=1}^{n}[f(w(t)) - f(w^*)]$$

$$\leq \mathfrak{R}_n + GR\frac{n}{B}\sum_{i=1}^{B}\phi_\xi(B + i)$$

$$+ \sqrt{\frac{2n}{B} \cdot \left(\frac{2}{3}\frac{GR}{B}\log\frac{4n}{\delta} + \sqrt{\frac{4}{9}\frac{G^2R^2}{B}(\log\frac{4n}{\delta})^2 + \left(4G^2R^2 + 16G^2R^2\sum_{i=1}^{B}\phi_\xi(i)\right)\log\frac{4n}{\delta}}\right) \cdot \log\frac{4\tau}{\delta}\log\frac{4n}{\delta}}.$$

*Proof.* From Proposition B.2, we obtain the following bound.

$$\sum_{t=1}^{n}[f(w(t)) - f(w^*)]$$

$$\leq \sum_{t=1}^{n}[f(w(t)) - F(w(t); x_{t+\tau-1}) + F(w^*; x_{t+\tau-1}) - f(w^*)] + \mathfrak{R}_n + G(\tau - 1)\sum_{t=1}^{n-\tau+1}\kappa(t) + GR(\tau - 1).$$

To complete the proof, it suffices to bound the first term; we define this term as

$$Z_n := \sum_{t=1}^{n}[f(w(t)) - F(w(t); x_{t+\tau-1}) + F(w^*; x_{t+\tau-1}) - f(w^*)].$$

We apply the same decomposition as the (13) of (Agarwal & Duchi, 2012). Define the index set $\mathcal{I}(i)$ as $\{1, \ldots, \lfloor\frac{n}{\tau}\rfloor + 1\}$ for $i \leq n - \tau\lfloor\frac{n}{\tau}\rfloor$ and $\{1, \ldots, \lfloor\frac{n}{\tau}\rfloor\}$ otherwise. Then we have

$$Z_n = \sum_{i=1}^{\tau}\sum_{t\in\mathcal{I}(i)}[X_t^i - \mathbb{E}[X_t^i|\mathcal{F}_{t-1}^i]] + \sum_{i=1}^{\tau}\sum_{t\in\mathcal{I}(i)}\mathbb{E}[X_t^i|\mathcal{F}_{t-1}^i],$$

where
$$X_t^i = f\big(w((t-1)\tau + 1)\big) - f(w^*) + F(w^*; x_{t\tau+i-1}) - F\big(w((t-1)\tau + 1); x_{t\tau+i-1}\big).$$

Note that by Lemma 4.1, we have that $\mathbb{E}[X_t^i | \mathcal{F}_{t-1}^i] \leq \frac{GR}{B} \sum_{i=1}^B \phi_\xi(\tau B + i)$. Then, we have

$$\mathbb{P}\Big(Z_n > \frac{nGR}{B} \sum_{i=1}^B \phi_\xi(\tau B + i) + \gamma\Big) \leq \mathbb{P}(\sum_{i=1}^\tau \sum_{t \in \mathcal{I}(i)} [X_t^i - \mathbb{E}[X_t^i | \mathcal{F}_{t-1}^i]] > \gamma)$$

$$\leq \mathbb{P}\Big(\bigcup_{i=1}^\tau \Big\{ \sum_{t \in \mathcal{I}(i)} [X_t^i - \mathbb{E}[X_t^i | \mathcal{F}_{t-1}^i]] > \frac{\gamma}{\tau} \Big\}\Big)$$

$$\leq \sum_{i=1}^\tau \mathbb{P}\Big( \sum_{t \in \mathcal{I}(i)} [X_t^i - \mathbb{E}[X_t^i | \mathcal{F}_{t-1}^i]] > \frac{\gamma}{\tau}\Big).$$

Define $Y := \sum_{t \in \mathcal{I}(i)} [X_t^i - \mathbb{E}[X_t^i | \mathcal{F}_{t-1}^i]]$ and $\alpha := \frac{\lambda}{\sqrt{B}}$. Notice that $X_t^i - \mathbb{E}[X_t^i | \mathcal{F}_{t-1}^i]$ is a centered random variable, that is, $\mathbb{E}[X_t^i - \mathbb{E}[X_t^i | \mathcal{F}_{t-1}^i]] = 0$. Then by the generalized Azuma's inequality (Lemma B.3), we conclude that

$$\mathbb{P}\Big(Y \geq \frac{\gamma}{\tau}\Big) \leq 2\exp\Big(-\frac{\gamma^2}{2\tau^2 \frac{n}{\tau} \alpha^2}\Big) + \sum_{t=1}^{\frac{n}{\tau}} \mathbb{P}(|X_t^i - \mathbb{E}[X_t^i | \mathcal{F}_{t-1}^i]| \geq \alpha).$$

The second term can be bounded by using the generalized Bernstein's inequality. The detailed calculation can be found in the discussion after Lemma B.4. We obtain that

$$\mathbb{P}\big(|X_t^i - \mathbb{E}[X_t^i | \mathcal{F}_{t-1}^i]| \geq \alpha\big) \leq 2\exp\Big(-\frac{\lambda^2}{C + \frac{4}{3}GR\frac{\lambda}{\sqrt{B}} + 16G^2R^2 \sum_{i=1}^B \phi_\xi(i)}\Big),$$

where $C = 4G^2R^2$. In summary, the concentration bound for $Z_n$ is

$$\mathbb{P}\Big(Z_n > GR\frac{n}{B} \sum_i \phi_\xi(\tau B + i) + \gamma\Big)$$

$$\leq 2\tau\exp\Big(-\frac{\gamma^2}{2\tau^2 \frac{n}{\tau} \alpha^2}\Big) + \tau \sum_{t=1}^{\frac{n}{\tau}} \mathbb{P}(|X_t^i - \mathbb{E}[X_t^i | \mathcal{F}_{t-1}^i]| \geq \alpha)$$

$$\leq 2\tau\exp\Big(-\frac{\gamma^2}{2\tau n \frac{\lambda^2}{B}}\Big) + 2n\exp\Big(-\frac{\lambda^2}{C + \frac{4}{3}GR\frac{\lambda}{\sqrt{B}} + 16G^2R^2 \sum_{i=1}^B \phi_\xi(i)}\Big).$$

Then, let $\frac{\delta}{2} = 2n\exp\big(-\frac{\lambda^2}{C + \frac{4}{3}GR\frac{\lambda}{\sqrt{B}} + 16G^2R^2 \sum_{i=1}^B \phi_\xi(i)}\big)$, and we obtain that

$$\lambda^2 = \Big(C + \frac{4}{3}GR\frac{\lambda}{\sqrt{B}} + 16G^2R^2 \sum_{i=1}^B \phi_\xi(i)\Big) \cdot \log \frac{4n}{\delta}.$$

It is a quadratic function of $\lambda$. Solving it yields that

$$\lambda = \frac{2}{3}\frac{GR}{B}\log\frac{4n}{\delta} + \sqrt{\frac{4}{9}\frac{G^2R^2}{B}\Big(\log\frac{4n}{\delta}\Big)^2 + \Big(C + 16G^2R^2 \sum_{i=1}^B \phi_\xi(i)\Big)\log\frac{4n}{\delta}}. \quad (13)$$

Also, let $\frac{\delta}{2} = 2\tau\exp\Big(-\frac{\gamma^2}{2\tau n \frac{\lambda^2}{B}}\Big)$, we have that

$$\gamma^2 = 2\tau n \frac{\lambda^2}{B} \cdot \log\frac{4\tau}{\delta}.$$

Substituting (13) into the above equation, we obtain that

$$\gamma = \sqrt{\frac{2\tau n}{B} \cdot \Big(\frac{2}{3}\frac{GR}{B}\log\frac{4n}{\delta} + \sqrt{\frac{4}{9}\frac{G^2R^2}{B}\Big(\log\frac{4n}{\delta}\Big)^2 + \Big(C + 16G^2R^2 \sum_{i=1}^B \phi_\xi(i)\Big)\log\frac{4n}{\delta}}\Big) \cdot \log\frac{4\tau}{\delta}\log\frac{4n}{\delta}}.$$

Then, we conclude that with probability at least $1 - \delta$,

$$\sum_{t=1}^{n}[f(w(t)) - f(w^*)]$$

$$\leq GR\frac{n}{B}\sum_{i=1}^{B}\phi_\xi(\tau B + i)$$

$$+ \sqrt{\frac{2\tau n}{B}\cdot\left(\frac{2}{3}\frac{GR}{B}\log\frac{4n}{\delta} + \sqrt{\frac{4}{9}\frac{G^2R^2}{B}\left(\log\frac{4n}{\delta}\right)^2 + \left(4G^2R^2 + 16G^2R^2\sum_{i=1}^{B}\phi_\xi(i)\right)\log\frac{4n}{\delta}}\right)\cdot\log\frac{4\tau}{\delta}\log\frac{4n}{\delta}}$$

$$+ \Re_n + G(\tau - 1)\sum_{t=1}^{n-\tau+1}\kappa(t) + GR(\tau - 1). \tag{14}$$

The desired result follows by noting that $\sum_{t=1}^{n} f(w(t)) \geq nf(\widehat{w}_n)$. $\qquad\square$

## C    REGRET ANALYSIS OF MINI-BATCH SGD

In this section, we derive the regret bound of mini-batch SGD algorithm. Throughout, for each sample loss $F(w;\xi)$, recall that its gradient $\|\nabla F(w;\xi)\|$ is uniformly bounded by $G$ (see Assumption 2.1). In particular, we assume the $k$-th coordinate of $\nabla F(w;\xi)$ is uniformly bounded by $G_k$, and we have $G = \sqrt{\sum_k G_k^2}$.

**1. Gradient Variance Bound under Dependent Data**

In the i.i.d. setting, the variance of stochastic gradient decreases as the batch size increases. Specifically, we have

$$\mathbb{E}\|\frac{1}{B}\sum_{i=1}^{B}\nabla F(w;\xi_i) - \nabla f(w)\|^2 = \frac{1}{B^2}\sum_{i=1}^{B}\mathbb{E}\|\nabla F(w;\xi_i) - \nabla f(w)\|^2 \leq \frac{2G^2}{B}.$$

Therefore, $\mathbb{E}\|\frac{1}{B}\sum_{i=1}^{B}\nabla F(w;\xi_i) - \nabla f(w)\|^2 = \mathcal{O}(\frac{1}{B})$. However, this bound no longer holds if the data samples are dependent. In the following lemma, we develop a similar result when the data is collected from a dependent stochastic process. Recall that $\nabla F(w(t);x_t)$ denotes the averaged gradient over the mini-batch $x_t$, i.e.,

$$\nabla F(w(t);x_t) = \frac{1}{B}\sum_{i=1}^{B}F(w(t);\xi_t^{(i)}).$$

**Lemma C.1.** *Let $\{w(t)\}_{t\in\mathbb{N}}$ be the model parameter sequence generated by the mini-batch SGD in (6). Let Assumptions 2.1 and 3.1 hold. Then, with probability at least $1 - \delta$,*

$$\|\nabla F(w(t);x_t) - \nabla f(w(t))\|^2 \leq \left[\frac{268}{3}G^2 + 256G^2\sum_{j=1}^{B}\phi_\xi(j)\right]\cdot\frac{\log\frac{2d}{\delta}}{B} + 2G^2\left(\frac{\sum_{i=1}^{B}\phi_\xi(i)}{B}\right)^2.$$

*Proof.* Let $x_t = \{\xi_t^{(i)}\}_{i=1}^{B}$ be the $t$-th mini-batch samples. We consider the filtration within $x_t$ and denote it as $\{\widehat{\mathcal{F}}_t^{(i)}\}$. Then, by the definition of canonical filtration,

$$X_i := \nabla F(w(t);\xi_t^{(i)})$$

is measurable with respect to $\widehat{\mathcal{F}}_t^{(i)}$. Define

$$Y_{i,k} := (X_i - \mathbb{E}[X_i|\mathcal{F}_{t-1}])_k$$

where $(\cdot)_k$ denotes the $k$-th entry of the specified vector. And it is easy to see that $\{Y_{i,k}\}_i$ is a centered process for any $k \in \{1, 2, \ldots, d\}$. With these construction, we start from the following

decomposition.

$$\|\nabla F(w(t); x_t) - \nabla f(w(t))\|^2$$
$$=\|\nabla F(w(t); x_t) - \mathbb{E}[\nabla F(w(t); x_t)|\mathcal{F}_{t-1}] + \mathbb{E}[\nabla F(w(t); x_t)|\mathcal{F}_{t-1}] - \nabla f(w(t))\|^2$$
$$\leq 2\underbrace{\|\nabla F(w(t); x_t) - \mathbb{E}[\nabla F(w(t); x_t)|\mathcal{F}_{t-1}]\|^2}_{(A)} + 2\underbrace{\|\mathbb{E}[\nabla F(w(t); x_t)|\mathcal{F}_{t-1}] - \nabla f(w(t))\|^2}_{(B)}.$$

Then we will bound the term (A) and (B), respectively.

- **Bounding (A):** Note that

$$\|\nabla F(w(t); x_t) - \mathbb{E}[\nabla F(w(t); x_t)|\mathcal{F}_{t-1}]\|^2 = \frac{1}{B^2}\|\sum_{i=1}^{B}[X_i - \mathbb{E}[X_i|\mathcal{F}_{t-1}]]\|^2$$

$$= \frac{1}{B^2}\sum_{k=1}^{d}\Big[\sum_{i=1}^{B}(X_i - \mathbb{E}[X_i|\mathcal{F}_{t-1}])_k\Big]^2$$

$$= \frac{1}{B^2}\sum_{k=1}^{d}\Big[\sum_{i=1}^{B}Y_{i,k}\Big]^2.$$

Then, we show that the process $\{Y_{i,k}\}_i$ satisfies the conditions of Lemma B.4.

- Since $\mathbb{E}[Y_{i,k}|\mathcal{F}_{t-1}] = 0$, we conclude that $\{Y_{i,k}\}_i$ is a centered process.
- Denote the $k$-th entry of $X_i$ as $X_{i,k}$. We know that $|X_{i,k}| \leq G_k$. Hence, we conclude that $0 \leq |Y_{i,k}| \leq 2G_k$. Then, we can set $b_i = 2G_k$ for all $i$.
- Lastly, we can bound the quantity $q$ defined in Lemma B.4 as follows.

$$q \leq 2G_k \sum_{j=1}^{B}\sum_{i=1}^{j-1}\|\mathbb{E}[Y_{j,k}|\mathcal{F}_t^{(i)}]\| + \frac{4}{3}G_k^2 B$$

$$\leq 4G_k^2 \sum_{j=1}^{B}\sum_{i=1}^{j-1}\phi_\xi(j-i) + \frac{4}{3}G_k^2 B$$

$$\leq 4G_k^2 B \sum_{j=1}^{B}\phi_\xi(j) + \frac{4}{3}G_k^2 B.$$

Now, we can apply Lemma B.4 and obtain that

$$\mathbb{P}\left(\sum_i Y_{i,k} > \lambda\right) \leq \exp\left(-\frac{\lambda^2}{\frac{134}{3}G_k^2 B + 128G_k^2 B \sum_{j=1}^{B}\phi_\xi(j)}\right).$$

With a union bound, we obtain that

$$\mathbb{P}\left(|\sum_i Y_{i,k}| > \lambda\right) \leq 2\exp\left(-\frac{\lambda^2}{\frac{134}{3}G_k^2 B + 128G_k^2 B \sum_{j=1}^{B}\phi_\xi(j)}\right).$$

Further applying the union bound over $k = 1, 2, \ldots, d$, we obtain that

$$\mathbb{P}\left(\bigcup_{k=1}^{d}\left\{|\sum_i Y_{i,k}|^2 > \lambda_k^2\right\}\right) \leq 2\sum_k \exp\left(-\frac{\lambda_k^2}{\frac{134}{3}G_k^2 B + 128G_k^2 B \sum_{j=1}^{B}\phi_\xi(j)}\right).$$

Let $\frac{\delta}{d} = 2\exp\left(-\frac{\lambda_k^2}{\frac{134}{3}G_k^2 B + 128G_k^2 B \sum_{j=1}^{B}\phi_\xi(j)}\right)$, we obtain that

$$\lambda_k^2 = \Big[\frac{134}{3}G_k^2 B + 128G_k^2 B \sum_{j=1}^{B}\phi_\xi(j)\Big] \cdot \log\frac{2d}{\delta}.$$

Then we conclude that,

$$\mathbb{P}\left(\bigcap_{k=1}^{d}\left\{|\sum_{i}Y_{i,k}|^2 \leq \left[\frac{134}{3}G_k^2 B + 128G_k^2 B \sum_{j=1}^{B}\phi_{\xi}(j)\right]\cdot\log\frac{2d}{\delta}\right\}\right) \geq 1 - \delta.$$

It implies that with the probability at least $1 - \delta$,

$$\sum_{k}|\sum_{i}Y_{i,k}|^2 \leq \left[\frac{134}{3}B\left(\sum_{k}G_k^2\right) + 128B\left(\sum_{j=1}^{B}\phi_{\xi}(j)\right)\left(\sum_{k}G_k^2\right)\right]\cdot\log\frac{2d}{\delta}.$$

By definition, $G = \sqrt{\sum_k G_k^2}$. Finally, we have the following bound for term (A): with probability at least $1 - \delta$,

$$\|\nabla F(w(t); x_t) - \mathbb{E}[\nabla F(w(t); x_t)|\mathcal{F}_{t-1}]\|^2 \leq \left[\frac{134}{3}G^2 + 128G^2\sum_{j=1}^{B}\phi_{\xi}(j)\right]\cdot\frac{\log\frac{2d}{\delta}}{B}.$$

- **Bounding (B):** Note that

$$\|\mathbb{E}[\nabla F(w(t); \xi_t^{(i)})|\mathcal{F}_{t-1}] - \nabla f(w(t))\| = \left\|\int\nabla F(w(t); \xi_t^{(i)})\mathrm{d}\mathbb{P}(\xi_t^{(i)} \in \cdot|\mathcal{F}_{t-1}) - \int\nabla F(w(t); \xi)\mathrm{d}\mu(\xi)\right\|$$

$$\leq \int\|\nabla F(w(t); \xi_t^{(i)})\||\mathrm{d}\mathbb{P}(\xi_t^{(i)} \in \cdot|\mathcal{F}_{t-1}) - \mathrm{d}\mu|$$

$$\leq G\cdot\phi_{\xi}(i).$$

Then we bound the norm by triangle inequality,

$$\|\mathbb{E}[\nabla F(w(t); x_t)|\mathcal{F}_{t-1}] - \nabla f(w(t))\| \leq \frac{1}{B}\sum_{i=1}^{B}\|\mathbb{E}[\nabla F(w(t); \xi_t^{(i)})|\mathcal{F}_{t-1}] - \nabla f(w(t))\|$$

$$\leq \frac{G}{B}\sum_{i=1}^{B}\phi_{\xi}(i).$$

Finally, we obtain the bound for the term (B) as

$$\|\mathbb{E}[\nabla F(w(t); x_t)|\mathcal{F}_{t-1}] - \nabla f(w(t))\|^2 \leq G^2\left(\frac{\sum_{i=1}^{B}\phi_{\xi}(i)}{B}\right)^2.$$

Combing the bounds of (A) and (B) yields that with probability at least $1 - \delta$,

$$\|\nabla F(w(t); x_t) - \nabla f(w(t))\|^2 \leq \left[\frac{268}{3}G^2 + 256G^2\sum_{j=1}^{B}\phi_{\xi}(j)\right]\cdot\frac{\log\frac{2d}{\delta}}{B} + 2G^2\left(\frac{\sum_{i=1}^{B}\phi_{\xi}(i)}{B}\right)^2.$$

$\square$

## 2. High-Probability Regret Bound

To derive the regret bound for the mini-batch SGD algorithm, we make the following additional mild assumption.

**Assumption C.2.** *The stochastic optimization problem (P) satisfies*

1. *Each sample loss $F(\cdot; \xi) : \mathcal{W} \to \mathbb{R}$ is convex.*

2. *The objective function $f : \mathcal{W} \to \mathbb{R}$ is L-smooth.*

**Theorem C.3** (High-probability regret bound). *Let $\{w(t)\}_{t\in\mathbb{N}}$ be the model parameter sequence generated by the mini-batch SGD in (6). Suppose Assumptions C.2, 3.1 and 2.1 hold. Then, with probability at least $1 - \delta$,*

$$\mathfrak{R}_T \leq \frac{\|w(1) - w^*\|^2}{2\eta} + \eta T\Big[\Big(\frac{268}{3}G^2 + 256G^2\sum_{j=1}^{B}\phi_\xi(j)\Big)\frac{\log\frac{2dT}{\delta}}{B} + 2G^2\Big(\frac{\sum_{i=1}^{B}\phi_\xi(i)}{B}\Big)^2\Big]$$

$$+ 2\eta L\sum_{t=1}^{T}\big(f(w(t)) - f(w^*)\big).$$

*Moreover, let $\eta = \mathcal{O}\big(\sqrt{\frac{B}{T\cdot\sum_{j=1}^{B}\phi_\xi(j)}}\big)$, the optimized upper bound is in the order of*

$$\mathfrak{R}_T = \widetilde{\mathcal{O}}\Big(\sqrt{\frac{T\cdot\sum_{j=1}^{B}\phi_\xi(j)}{B}}\Big) + 2\eta L\sum_{t=1}^{T}\big(f(w(t)) - f(w^*)\big).$$

*Proof.* For convenience, we define $g_t = \frac{1}{B}\sum_{i=1}^{B}\nabla F(w(t); \xi_t^{(i)})$. By the algorithm update (6), we obtain that

$$2\langle g_t, w(t) - w^*\rangle \leq \frac{\|w(t) - w^*\|^2 - \|w(t+1) - w^*\|^2}{\eta} + \eta\|g_t\|^2$$

$$\leq \frac{\|w(t) - w^*\|^2 - \|w(t+1) - w^*\|^2}{\eta} + 2\eta\|g_t - \nabla f(w(t))\|^2 + 2\eta\|\nabla f(w(t))\|^2.$$

Summing the above inequality over $t$ yields that

$$2\sum_{t=1}^{T}\langle g_t, w(t) - w^*\rangle$$

$$\leq \frac{\|w(1) - w^*\|^2 - \|w(T+1) - w^*\|^2}{\eta} + 2\eta\sum_{t=1}^{T}\|g_t - \nabla f(w(t))\|^2 + 4\eta L\sum_{t=1}^{T}(f(w(t)) - f(w^*)).$$

By convexity of the function, we further obtain that

$$2\sum_{t=1}^{T}(F(w(t); x_t) - F(w^*; x_t)) \leq \frac{\|w(1) - w^*\|^2}{\eta} + 2\eta\sum_{t=1}^{T}\|g_t - \nabla f(w(t))\|^2 + 4\eta L\sum_{t=1}^{T}(f(w(t)) - f(w^*)).$$

Then, we apply Lemma C.1 to bound the second term $\sum_{t=1}^{T}\|g_t - \nabla f(w(t))\|^2$ and then apply a union bound on over $t$. We conclude that, with probability at least $1 - \delta$,

$$\sum_{t=1}^{T}(F(w(t); x_t) - F(w^*; x_t))$$

$$\leq \frac{\|w(1) - w^*\|^2}{2\eta} + \eta T\cdot\Big[\Big(\frac{268}{3}G^2 + 256G^2\sum_{j=1}^{B}\phi_\xi(j)\Big)\frac{\log\frac{2dT}{\delta}}{B} + 2G^2\Big(\frac{\sum_{i=1}^{B}\phi_\xi(i)}{B}\Big)^2\Big]$$

$$+ 2\eta L\sum_{t=1}^{T}(f(w(t)) - f(w^*)).$$

The proof is completed. Lastly, we set the learning rate $\eta$. To minimize the obtained upper bound, it suffices to minimize the first two terms, as the last term can be combined with the left hand side of (14) when we apply this regret bound. The optimized learning rate is achieved when

$$\frac{\|w(1) - w^*\|^2}{2\eta} = \eta T\cdot\Big[\Big(\frac{268}{3}G^2 + 256G^2\sum_{j=1}^{B}\phi_\xi(j)\Big)\frac{\log\frac{2dT}{\delta}}{B} + 2G^2\Big(\frac{\sum_{i=1}^{B}\phi_\xi(i)}{B}\Big)^2\Big].$$

Then, $\eta$ is chosen as

$$\eta = \sqrt{\frac{\|w(1) - w^*\|^2/2}{T \cdot \left[\left(\frac{268}{3}G^2 + 256G^2 \sum_{j=1}^{B} \phi_\xi(j)\right)\frac{\log \frac{2dT}{\delta}}{B} + 2G^2 \left(\frac{\sum_{i=1}^{B} \phi_\xi(i)}{B}\right)^2\right]}}$$

$$= \mathcal{O}\left(\sqrt{\frac{B}{T \cdot \sum_{j=1}^{B} \phi_\xi(j)}}\right).$$

$\square$

## D    EXPERIMENT SETUP

Recall that we consider the following convex quadratic optimization problem:

$$\min_{w \in \mathbb{R}^d} \mathbb{E}_{\xi \sim \mu}(w - \xi)^T A(w - \xi),$$

where $A$ is a fixed positive semi-definite matrix and $\mu$ is the uniform distribution on $[0,1]^d$. The data stream admitting such a stationary distribution $\mu$ can be generated by a certain Metropolis-Hastings sampler provided in (Jarner & Roberts, 2002). Specifically, it is described as follows.

**Step 1:** Let the "proposal" distribution $q(x)$ have the density of Beta$(r + 1, 1)$; that is,

$$q(x) = \begin{cases} (r+1)x^r & x \in [0,1] \\ 0 & x \notin [0,1] \end{cases}.$$

Define the acceptance probability $\alpha(x, y) = \min\{\frac{q(x)}{q(y)}, 1\}$.

**Step 2:** If the current state is $\xi_t$, then we sample $\zeta \sim q$. Define the next state $\xi_{t+1}$:

$$\xi_{t+1} = \begin{cases} \xi_t & \text{w.p. } 1 - \alpha(\xi_t, \zeta), \\ \zeta & \text{w.p. } \alpha(\xi_t, \zeta). \end{cases}$$

**Step 3:** Go back to **Step 2** to generate the next state.

We repeatedly generate $d$ independent sequences starting from the same initial state $s_0 = 0$ to obtain a $d$-dimension Markov chain. It has been shown that the above generated Markov chain converges to $\mu$ in distribution with an algebraic convergence rate $\phi_\xi(k) \leq \mathcal{O}(k^{-1/r})$ in Proposition 5.2, (Jarner & Roberts, 2002).

We consider the following bias term at the fixed point $w = \mathbf{0}_d$.

(Bias):    $\left|\mathbb{E}\left[F(w; x_\tau)|x_0 = \mathbf{0}_d\right] - f(w)\right|.$

It can be used to approximate the left-hand side of Lemma 4.1. Since $\mathbb{E}\left[F(w; x_\tau)|s_0 = \mathbf{0}_d\right]$ cannot be explicitly obtained, we use Monte Carlo method to estimate this conditional expectation. That is, we generate $n = 10,000$ independent trajectories starting from $x_0 = \mathbf{0}_d$. At the step $\tau$, we estimate the expected value as $\frac{1}{n}\sum_{i=1}^{n} F(w; x_\tau^{(i)})$, where $x_\tau^{(i)}$ with the superscript $(i)$ indicates that it is sampled from the $i$-th trajectory. Then we investigate the relation between the step $\tau$ and the mixing parameter $r$ and the relation between the step $\tau$ and the batch size $B$. All the results are presented in Section 5.

