# OpenReview forum: "How to Improve Sample Complexity of SGD over Highly Dependent Data?"
_ICLR.cc/2022/Conference — ICLR 2022 Submitted_

### Official Review · Reviewer_PELk · 2021-10-26

**Correctness:** 3
**Technical Novelty And Significance:** 2
**Empirical Novelty And Significance:** 2
**Recommendation:** 5
**Confidence:** 3

**Main Review:**

This paper proves that the subsampling method can reduce the samples complexity of SGD when data are dependent.

This paper has some merits.

1. Compared with existing works [1,2], the paper considers more general mixing time, that is, algebraic mixing time (fast and slow). The authors prove that under the algebraic mixing time case,   data-subsampling can improve the   sample complexity more efficient than the
geometric case.

2. Due to the subsampling, the authors prove a more general regret result in Theorem 4.2.

However, some weaknesses hurt this paper.

1. The contribution is incremental due to existing works [1,2]. This paper does not present any novel method or solve theoretical problems. The subsampling method has been presented in [1,2]. And they also show how the samples are reduced by the subsampling method. The only novel part is about the proof in geometric case but is very easy by following existing proofs in [1,2].

2. While the regret for data-dependent SGD is given in [3,4]. Although subsampling modifies the traditional sampling method. The proof is easy if we use the techniques from [3,4].


Minor comments:
[3] has been published in SIAM J. Optimization. The journal version contains more details. I suggest the authors update the reference in the future version.

[1] Least Squares Regression with Markovian Data: Fundamental Limits and Algorithms.
[2] SIMPLE AND OPTIMAL METHODS FOR STOCHASTIC VARIATIONAL INEQUALITIES, II: MARKOVIAN NOISE AND POLICY EVALUATION IN REINFORCEMENT LEARNING
[3] Ergodic subgradient descent.
[4] The generalization ability of online algorithms for dependent data


**Summary Of The Paper:**

This paper proves the subsampling method can reduce the samples complexity of SGD on dependent data when the mixing time follows more general distributions.

**Summary Of The Review:**

Due to existing works in this area, I do not think the current version of this paper is beyond the publication bar.

---

### Official Review · Reviewer_RcE4 · 2021-11-01

**Correctness:** 3
**Technical Novelty And Significance:** 1
**Empirical Novelty And Significance:** 2
**Recommendation:** 3
**Confidence:** 5

**Main Review:**

Strength:

Analyzes minibatch SGD for phi-mixing data.

Weakness:

Since this is a theoretical paper I am basing my impression of the paper on its theoretical novelty.

The result related to periodically subsampled SGD is a minor modification of Theorem 2 of (Agarwal & Duchi, 2012). There is no novelty here.

The result related to minibatch SGD: For the proof of Theorem 4.2 there are three main components: Lemma 4.1,  applying Bernstein's inequality for phi-mixing, generalized Azuma's inequality to deal with the bias. Lemma 4.1  almost follows immediately from the definition of $\phi$-mixing coefficients. The Bernstein's inequality for dependent sequence was obtained in Deylon et. al. 2009 and so it's not a contribution of this paper. The authors claim that they generalize Proposition 34 of Tao et.al. 2015 to get the generalized Azuma's inequality. It is not clear to me in which aspect it is generalized from Tao. If the authors could point out how it is more general that would be great.

**Summary Of The Paper:**

This work analyzes and provides high probability convergence guarantee for periodically subsampled and minibatch SGD for optimization of convex and Lipschitz continuous function when the data stream is $\phi$-mixing.

**Summary Of The Review:**

Overall, there is almost no theoretical novelty. So I am voting for reject.

---

### Official Review · Reviewer_J2CF · 2021-11-02

**Correctness:** 4
**Technical Novelty And Significance:** 3
**Empirical Novelty And Significance:** 3
**Recommendation:** 6
**Confidence:** 4

**Main Review:**

Strength-

* The analysis of SGD over dependent data, modeled by phi-mixing process, provides interesting insights on the performance of SGD for dependent data: the proposed framework provides theoretical evidence of how the convergence of SGD in the data-dependent setting is affected by the structure of the stochastic update scheme.

* The paper is well-organized and the message is clearly conveyed: this paper clearly conveys the message of why mini-bath SGD and SGD with subsampling can provide better convergence rate compared to SGD for dependent data based on the concrete analysis of sample complexity. Also, both theoretical result and empirical result demonstrate that the bias of SGD decreases as the subsampling period r and bath size B increases, respectively.

* Technical novelty compared to (Agarwal&Duchi 2012) is well summarized in the end of Sec. 4, even though the main framework to analyze the dependency structure, phi-mixing model, as well as the analysis for original SGD (Section 3.1) are directly addressed from (Agarwal&Duchi 2012).


Weakness-

* Tightness of the bound: Since the provided bounds are only upper bounds, it will be more interesting if the authors can provide any arguments or analysis about the tightness of the bounds, or lower bounds on sample complexity.

* Empirical results: it will be interesting to compare the empirical convergence rate of variants of SGD with the theoretical bounds provided by the authors to see the tightness of the bounds in non-asymptotic regime.


**Summary Of The Paper:**

This paper studies the sample complexity of a few variants of SGD in solving optimization problems over dependent data. The dependent data introduces non-negligible bias in SGD that slows down convergence of the algorithm, and this paper adopts \phi-mixing data dependence models, to quantify the level of dependence in the queried samples, and analyze how much the convergence of SGD can be slowed down in terms of the phi-mixing model. Then, this paper demonstrates the benefits of SGD with subsampling and mini-batch SGD over SGD for dependent data.

**Summary Of The Review:**

This paper provides a concrete framework to analyze the convergence rate of SGD over dependent data. By addressing the phi-mixing process to model the dependent data, the convergence rate of variants of SGD are compared and the benefits of mini-batch SGD over SGD for dependent data are clearly demonstrated from the analysis. This paper includes interesting insights and the addressed framework will be helpful to analyze SGD for other stochastic optimization problems using dependent data.

---

### Official Review · Reviewer_xaGe · 2021-11-03

**Correctness:** 4
**Technical Novelty And Significance:** 2
**Empirical Novelty And Significance:** 2
**Recommendation:** 5
**Confidence:** 4

**Main Review:**

## Convergence results
The paper is clearly written and provides a good summary of the existing work. It also elucidates many real-world examples, which motivate the setting of dependent noise structure in stochastic optimization. However, I feel that the paper doesn't have many new results and is unable to generalize existing results sufficiently.

Until section 3.2 no new result has been presented. And the result with sub-sampling in section 3.2 is a trivial extension of the existing high probability guarantee by Agrawal and Duchi. I agree that the current writing makes it very clear what the contribution of this paper is, but I feel like it is redundant and not appropriate for a 9-page submission. It would have made much more sense for the paper to actually focus on the new techniques developed for proving results for MB-SGD. These details are currently buried in the appendix, and only briefly mentioned at the end of section 4. Having said that, the convergence result for MB-SGD is novel, and the discussion leading up to it gives a good intuition for why mini-batching would reduce the bias.

## Assumptions

I am also concerned about the lipschitzness assumption in 2.1.1. This essentially implies a bounded stochastic gradient condition, which is not even satisfied for simple losses such as least squares. In fact, it is not even satisfied for the instantaneous loss in the experiments of this paper. It would have been a significant improvement if the results of Agrawal and Duchi were extended to smooth functions with bounded variance (at the optimal), which would cover a much larger class of functions.

## Experiments

The experiments could have been done much more extensively as well. The paper mentions how different applications in reinforcement learning, etc. actually satisfy different $\phi$-mixing assumptions. It would have been interesting to see some experiments on real-world data which elucidated this. Moreover, it seems in the current formulation only the global losses need to be convex and the instantaneous losses can be non-convex. I wonder if that opens avenues for more interesting experiments as well. In the current figure 2, the colors, as well as the scaling, are really bad.

***
Overall, I believe the fundamental question which the paper asks,
> How does the structure of stochastic updates affect the convergence rate and sample complexity of stochastic algorithms over-dependent data?

is very interesting. But the paper makes very small steps towards answering this satisfactorily.



**Summary Of The Paper:**

In stochastic convex optimization (SCO) characterized by the following objective for convex $f$, $$\min_w f(w) := \mathbb{E}_{\xi\sim \mu}[F(w; \xi)],$$ we typically assume repeated access to the noise distribution $\mu$, for sampling i.i.d. random variables $(\xi_t)_t$. This access can be used to obtain unbiased estimates $\nabla F(w; \xi_t)$ of the true gradient $\nabla f(w)$. This oracle access can be used to implement stochastic gradient methods, which are well studied and known to have optimal sample complexity under certain regularity conditions.

On the other hand in online convex optimization, at each time step, we obtain a loss function $f_t$ from some function class $\mathcal{F}$, and we have to minimize some notion of regret. This can be modeled like SCO by defining, $f_t:= F(.; \xi_t)$, but since there was no restriction on $f_t$'s, the $\xi_t$'s here can have arbitrary dependencies over time. In particular, they could be chosen adversarially.

In between both these extremes is the setting where $\xi_t$ could have some dependency structure. One such structure called $\phi$-mixing was introduced by [Agarwal and Duchi](https://arxiv.org/abs/1110.2529). Under this dependence assumption, they provide upper bounds on the convergence of SGD. This paper generalizes these upper bounds to a sub-sampling variant of SGD as well as mini-batch SGD. It shows that mini-batch SGD essentially attains the $1/\epsilon^2$ sample complexity of for i.i.d. sampling when there is a low level of dependence in $\xi_t$'s, and more generally improves over vanilla SGD across various levels of dependence captured by the $\phi$-mixing assumption.

**Summary Of The Review:**

I think the paper is well written, clear, and has technical novelty. However, there aren't enough new results, to justify acceptance.

---

### Decision · Program_Chairs · 2022-01-20

**Decision:**

Reject

**Comment:**

Dear Authors,

This paper eventually received mostly negative reviews (scores 5, 3, 5), with one mildly positive review (score 6). All reviews were particularly informative, offering detailed and expert feedback. I was hoping for author engagement, but unfortunately, no rebuttal was submitted.

In general, the reviewers and me found the paper well written, on a timely topic, but of a very limited theoretical novelty. Well-articulated details of this can be found in the reviews and I would recommend the authors to consider them carefully in their revision. I have no option but to reject this work.

The main reason for rejection in this case is therefore limited theoretical novelty. However, this is a solid paper that is of publishable quality, albeit perhaps in a somehow lesser venue, at least in its current form.

Kind regards,

Area Chair